# Giant optomechanical spring effect in plasmonic nano- and picocavities probed by surface-enhanced Raman scattering

Lukas A. Jakob[1], William M. Deacon[1], Yuan Zhang [2] ✉, Bart de Nijs[1], Elena Pavlenko[1], Shu Hu[1], Cloudy Carnegie[1], Tomas Neuman[3], Ruben Esteban[3], Javier Aizpurua[3] ✉ & Jeremy J. Baumberg[1] ✉

Molecular vibrations couple to visible light only weakly, have small mutual interactions, and hence are often ignored for non-linear optics. Here we show the extreme confinement provided by plasmonic nano- and pico-cavities can sufficiently enhance optomechanical coupling so that intense laser illumination drastically softens the molecular bonds. This optomechanical pumping regime produces strong distortions of the Raman vibrational spectrum related to giant vibrational frequency shifts from an optical spring effect which is hundred-fold larger than in traditional cavities. The theoretical simulations accounting for the multimodal nanocavity response and near-field-induced collective phonon interactions are consistent with the experimentally-observed non-linear behavior exhibited in the Raman spectra of nanoparticle-on-mirror constructs illuminated by ultrafast laser pulses. Further, we show indications that plasmonic picocavities allow us to access the optical spring effect in single molecules with continuous illumination. Driving the collective phonon in the nanocavity paves the way to control reversible bond softening, as well as irreversible chemistry.

Molecular vibrations increasingly dominate electronic, thermal, and spin transport in a wide range of devices from photovoltaics[1–4] to molecular electronics[5] as well as being of fundamental interest. Vibrations also underpin label-free molecular sensing[6], harnessed with metal-induced plasmonic enhancements to overcome small Raman cross sections. Surface-Enhanced Raman Spectroscopy (SERS) is well-established for studying molecular vibrations[7,8], exciting the molecular ground state to the first vibrational level simultaneously with emission of a Stokes-shifted longer-wavelength photon. To enhance SERS signals, plasmonic nanostructures are designed to maximize the nanocavity optical field confinement in intense localized hot-spots in which molecules are immersed[9].

Recently it was shown that SERS can be described as molecular optomechanics, in which molecular vibrations and the optical nanocavity are highly coupled[10,11]. Despite their large nanocavity linewidths ($\kappa$), the ~200 nm$^3$ effective mode volumes $V$ yield single-plasmon optomechanical couplings $g$ exceeding 3 meV, approaching room temperature thermal energies[12–14]. So far, optomechanical models for plasmonic cavities used descriptions based on cavity-QED, extended to account for plasmonic losses, and were often restricted to a single resonant photonic mode[13]. The Stokes scattering spectrum, $S(\omega_\nu) \propto g^2(1+n_\nu)I_l$ at Raman shift $\omega_\nu$ did not vary in shape with pump laser intensity $I_l$ for excited vibrational population $n_\nu$.

[1]Nanophotonics Centre, Cavendish Laboratory, University of Cambridge, Cambridge CB3 0HE, UK. [2]Henan Key Laboratory of Diamond Optoelectronic Materials and Devices, Key Laboratory of Material Physics, Ministry of Education, School of Physics and Microelectronics, Zhengzhou University, Zhengzhou 450052, China. [3]Center for Material Physics (CSIC—UPV/EHU and DIPC), Paseo Manuel de Lardizabal 5, Donostia-San Sebastian Gipuzkoa 20018, Spain. ✉e-mail: yzhuaudipc@zzu.edu.cn; aizpurua@ehu.eus; jjb12@cam.ac.uk

Here, we show that this approximation is incomplete, and that a full multimodal treatment of the nanocavity[15] is needed to explain optical field-dependent softening of molecular vibrations seen when collectively driving molecules at higher powers. For the first time to the best of our knowledge, we report indications of a vibrational frequency shift associated with the optical spring effect, a novel effect in the context of molecular nanotechnology. The considerations raised here will also be important for optomechanics of phonons in thin crystals when integrated into plasmonic nanocavities, such as perovskites or 2D layered materials.

## Results

### Continuum-field description of plasmonic nanocavity

We start by theoretically considering a realistic plasmonic nanocavity which optimizes molecular optomechanical coupling. The molecules are embedded in a metal-insulator-metal nm-thick waveguide. This is easily realized using the nanoparticle-on-mirror geometry (NPoM), where a self-assembled monolayer (SAM) of active molecules is formed on a flat Au substrate before deposition of Au nanoparticles (NPs) on top (Fig. 1a)[16]. This places a few hundred molecules in the ~10 nm-wide optical field, highly-confined within the nanogap (thickness $d \sim 1.3$ nm) between the flat bottom facet of the 80 nm-diameter Au nanoparticle and the Au surface underneath (Fig. 1b). More details on the simulated nanostructure are given in Supplementary Note S3. The resulting optical field enhancement factor EF > 300 gives SERS $\propto$ EF$^4$ which is hence increased by $\geq 10^{10}$ or more[9,16].

Previous optomechanical models typically considered a dominant single cavity mode. However, if the full nanocavity plasmonic spectrum (grey line in Fig. 1c) with a more complex structure of resonances is included in a continuum-field model[15,17], new behaviours are predicted as compared to the single-mode description. Furthermore, within a general scheme of optomechanical dynamics, increasing the

optical pumping predicts spectral changes of the Stokes scattering due to vibrational shifts induced by optomechanical interactions with the plasmonic modes. These interactions are enhanced in molecular self-assemblies by the coupling of different molecules leading to collective phonon modes[18]. The effects above can be obtained from the dynamics of the vibrational amplitude $\beta_s$ of the $v^{th}$ vibrational mode of the $s^{th}$ molecule:

$$\frac{\partial}{\partial t}\beta_s = -i\left[(\omega_\nu - \mathrm{Re}\,v_{ss}) - i\left(\frac{\gamma_\nu}{2} + \mathrm{Im}\,v_{ss}\right)\right]\beta_s + i\sum_{s'\neq s}v_{s's}\beta_{s'} \quad (1)$$

with $\omega_\nu$, $\gamma_\nu$ its vibrational frequency and decay rate, respectively. We observe that both the frequency and decay rate are modified by the term $v_{s's} = (S^+_{ss'})^* + S^-_{ss'}$, given by the spectral density associated with the Stokes $(S^+_{ss'})^*$ and the anti-Stokes $S^-_{ss'}$ scattering, defined as (see Supplementary Note S1.2)

$$S^\pm_{ss'} = \frac{1}{4\hbar\varepsilon_0}\left(\frac{\omega_l\mp\omega_\nu}{c}\right)^2 [\mathbf{p}_s(\omega_l)]^* \cdot \overset{\leftrightarrow}{G}(\mathbf{r}_s,\mathbf{r}_{s'};\omega_l\mp\omega_\nu)\cdot\mathbf{p}_{s'}(\omega_l), \quad (2)$$

where $\hbar, \varepsilon_0, c$ are the reduced Planck constant, vacuum permittivity and speed of light. Here $\mathbf{p}_s(\omega_l) = \overset{\leftrightarrow}{\alpha}_\nu \mathbf{E}(\mathbf{r}_s, \omega_l)$ is the Raman dipole of the $s^{th}$ molecule induced by the local electric field $\mathbf{E}(\mathbf{r}_s, \omega_l)$, excited by a laser of frequency $\omega_l$, acting on the molecule at position $\mathbf{r}_s$ with Raman polarizability of the $v^{th}$ vibrational mode $\overset{\leftrightarrow}{\alpha}_\nu$. As observed in Eq. (2), in addition to the local field enhancement, the Green's function of the plasmonic system, $\overset{\leftrightarrow}{G}(\mathbf{r}_s,\mathbf{r}_{s'};\omega_l\mp\omega_\nu)$, at the Stokes, $\omega_l-\omega_\nu$, and anti-Stokes, $\omega_l+\omega_\nu$, frequencies is the key magnitude that governs the optomechanical interaction. The term $v_{s's}$ with $s = s'$ is explicitly separated in Eq. (1) and describes the self-interaction of the Raman-induced dipoles for a single molecule, where the imaginary part $\mathrm{Im}\{v_{ss}\}$ leads to an increase of the decay rate (broadening the Raman lines by $2\mathrm{Im}\{v_{ss}\} \propto \mathrm{Im}\{G\}$, as reported in previous work[13,15], while the real

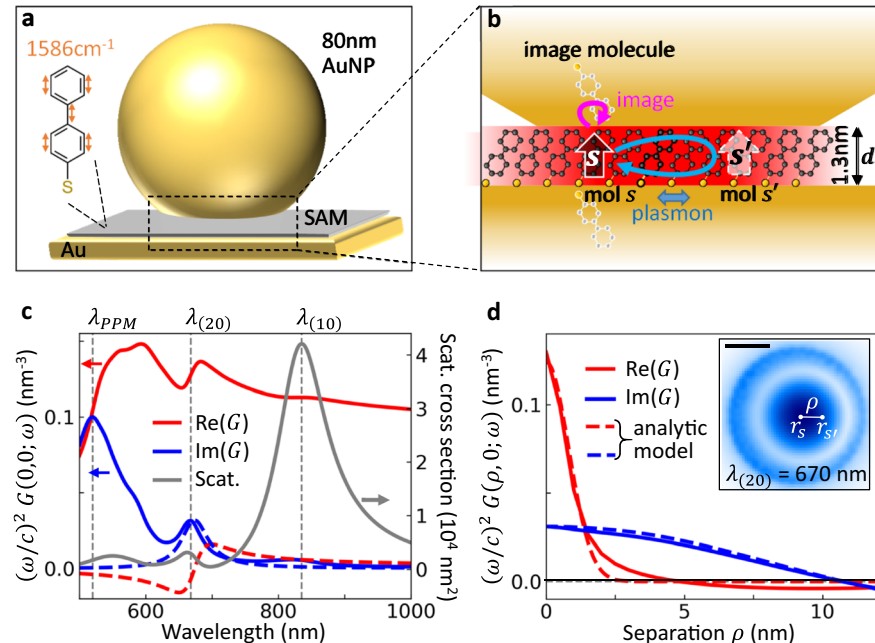

**Fig. 1 | Theory of nonlinear vibrational coupling in plasmonic nanocavities.**
**a** Schematic of 80 nm nanoparticle-on-mirror (NPoM) containing 1.3 nm-thick self-assembled monolayer (SAM) of biphenyl-4-thiol (BPT) molecules, showing benzene ring stretch at 1586 cm$^{-1}$. **b** Nanogap supports localized plasmon modes (red). BPT molecular Raman dipoles ($s$, $s'$) interact via their image molecules (pink arrow) and through plasmon modes (blue arrow). **c** Complex self-interaction Green's function (Re{$G$} = red, Im{$G$} = blue) for response produced by a vertical dipole in the gap centre. Scattering cross section (grey) shows dominant localized plasmonic (10)

(lowest order) and (20) (second order) modes (vertical dashed) at $\lambda_{(10)} \approx 830$ nm, $\lambda_{(20)} \approx 670$ nm, and a peak in Im$G$ at $\lambda_{PPM} \approx 520$ nm identified as the plasmon pseudo-mode (PPM), originating from overlapping higher order modes. Dashed curves show $G$ when only a single-optical-mode is considered in the model. **d** Two-point Green's function between spatially separated locations in the gap (separation $\rho$), at $\lambda_{(20)} = 670$ nm as obtained numerically (solid line) and with an analytic model based on image dipoles (dashed lines, see Supplementary Note S3.1). Inset shows midgap electric near-field at $\lambda_{(20)}$, with a scale bar 10 nm.

part, Re$\{v_{ss}\}$ leads to a reduction of the vibrational frequency of the mode (spectral shift of Raman lines, $\Delta\omega_\nu \propto \mathrm{Re}\{G\}$), corresponding to the optical spring effect in cavity optomechanics. In the case of many molecules, the terms $v_{s's}$ with $s \neq s'$ couple the different molecules, and the resulting collective response modifies the optomechanical effects, as explained below. In this work we explore the properties of and the evidence for this optical spring effect in our NPoM configuration. We note that the occurrence and magnitude of the observed effect is linked to the exact design of this nanostructure and its plasmonic cavity modes.

The self-interaction Green's function at the centre of the NPoM gap shows the relevant landscape of plasmonic modes (Fig. 1c). We focus on the contribution due to scattering by the NPoM (ignoring the contribution of the direct dipole-dipole interaction in the homogeneous medium, which avoids a divergence in the case of the self-interaction, and requires careful renormalization[19–21]). Importantly, Re$\{G\}$ for the full NPoM cavity is 10-fold larger than when a single-mode plasmonic cavity is considered (solid versus dashed red line, see Supplementary Note S6 for a discussion of the single-mode model), highlighting the importance of fully incorporating all plasmonic modes to correctly address the optomechanics. The spatial distribution of plasmonic modes impacts differently the real and imaginary parts of $G$ (Fig. 1d). While Im$\{v_{ss'}\} \propto \mathrm{Im}\{G(\rho = |\mathbf{r}_{s'} - \mathbf{r}_s|)\}$ (blue solid line in Fig. 1d) extends across the whole facet and follows the near-field of the (20) mode at $\lambda_{(20)} = 670$ nm (blue dashed in Fig. 1d), Re$\{v_{ss'}\} \propto \mathrm{Re}\{G(\rho)\}$ is seen to be extremely short-ranged due to the interaction of highly localized dipole image charges in the gap (Fig. 1d, red line). This short-range interaction is important for the vibrational shifts of many molecules, is nearly spectrally independent (Fig. 1c, red solid line), and can be analytically derived from the coupling of image dipoles in the gap (Fig. 1d red dashed line, see Supplementary Note S3.1), giving a profile of width $\delta \sim 0.9\, d$ for gap size $d$.

## Optical spring effect and collective phonon modes

We examine first the vibrational shift experienced by a single molecule at the centre of the gap. The real part of the spectral density, Re$\{S_{ss}^\pm\}$, associated with the Stokes (+), and anti-Stokes (-) frequencies, determines the total frequency shift of each vibrational mode. The resulting values of Re$\{S_{ss}^\pm\}$ are shown in Fig. 2a. Compared

to the single-mode model (dashed lines), the values of Re$\{S_{ss}^\pm\}$ are about ten-fold larger when considering the full plasmonic response (solid lines) because of the larger value of the real part of the Green's function (Fig. 1c). Further, in the single-mode model Re$\{S_{ss}^\pm\}$ both approach a maximum around the single plasmon cavity resonance (set to $\lambda_{(20)}$, see Supplementary Note S6) but with opposite sign. This results in a small total frequency shift from the optomechanical optical spring effect

$$\Delta_{\mathrm{os}}^1 = \mathrm{Re}\{S_{ss}^+ + S_{ss}^-\} \tag{3}$$

due to cancelation of frequency shifts from combining both Stokes and anti-Stokes contributions. In contrast, in the full plasmonic model $S_{ss}^\pm$ both approach a maximum around plasmon modes but have the same sign, which results in a much larger total frequency shift than in the single cavity mode description. We therefore predict a substantial optical spring effect in our NPoM configuration.

This significant optical spring effect in molecular self-assemblies is further collectively enhanced by coupling of molecules laterally separated by $\rho = |\mathbf{r}_{s'} - \mathbf{r}_s|$ arising from their image-dipole local Coulomb interactions. The term $v_{ss'}$ with $s \neq s'$ in Eq. (1) describes the interaction between the Raman-induced dipoles, which leads to formation of collective phonon modes by superposing vibrations in individual molecules. In the mid-infrared, such collective phonons have been termed vibrational excitons[2,22–24], but here they are dynamically induced only by the laser driven Raman dipoles. We suggest the term "vibrational exciton" in the literature is misleading since excitons refer to bound electron-hole pairs, while the collective vibrations here are analogous to localized phonons in a continuous material. We thus prefer the term "molecular phonons". This collective response results in the emergence of Raman-bright collective phonons across all $N_m$ molecules (which fit inside the NP facet and interact in the NPoM gap), together with other dark collective modes. The optical spring shift induced by the fundamental bright collective mode corresponds to a bond softening, and scales linearly with $N_m$ (Fig. 2b). This dependency is obtained by calculating the eigenmodes of Eq. (1) for an increasing number of molecules (see Supplementary Note S5.4) all with vibrational frequency $\omega_\nu = \omega_{\nu 1} = 1586$ cm$^{-1}$. The shift for 100 molecules forming a dense square patch at the centre of the NPoM gap (inset Fig. 2c) is 32 times

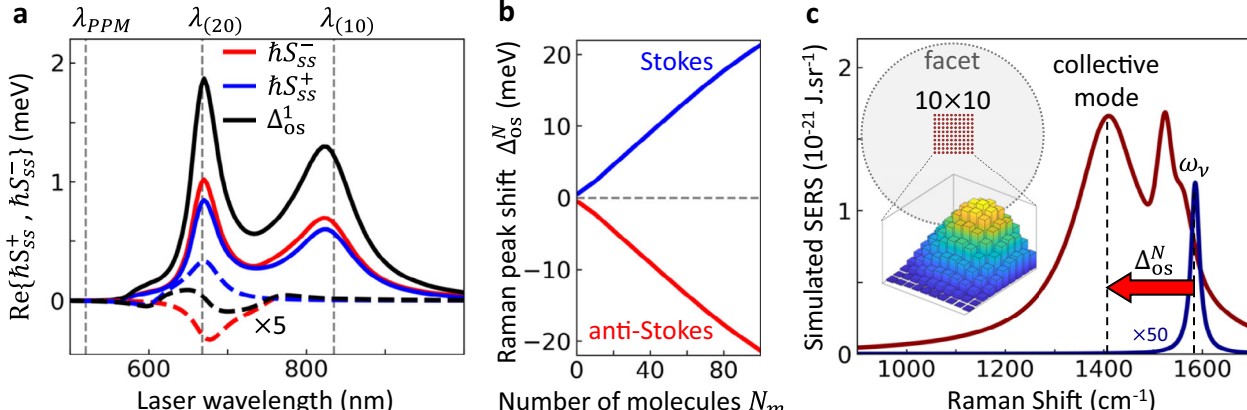

**Fig. 2 | Origin of optical spring effect in molecular optomechanics. a** Optical spring effect *vs* laser wavelength for $\omega_{\nu 1} = 1586$ cm$^{-1}$ mode in the NPoM gap, showing contributions of the Stokes $S^+$ (blue) and anti-Stokes $S^-$ (red) optomechanical parameters to the total vibrational shift, $\Delta_{\mathrm{os}}^1$, for a single molecule in the nanocavity centre. $S^+$ and $S^-$ scale linearly with laser intensity (shown here for $10^7$ µWµm$^{-2}$). Dashed curves show single-mode plasmonic cavity results. **b** Dependence of shift in the fundamental collective Raman bright mode $\Delta_{\mathrm{os}}^N$ with the number of molecules $N_m$ arranged in a lattice at the middle of gap (Stokes in

blue, anti-Stokes in red) at $5 \times 10^7$ µWµm$^{-2}$. **c** SERS emission from full multi-molecule model for the $\omega_{\nu 1} = 1586$ cm$^{-1}$ mode with 633 nm CW pump intensity of $10^5$ µWµm$^{-2}$ (blue, multiplied by 50) and $5 \times 10^7$ µW µm$^{-2}$ (red). At the larger intensity the broad peak is down-shifted $\Delta_{\mathrm{os}}^N \sim 170$ cm$^{-1}$ from $\omega_\nu$ due to the dominant bright Raman collective phonon mode. Top inset shows the square array of 100 molecules spaced $\rho = 0.58$ nm apart and centred in the facet (dashed, radius 16 nm). Bottom inset shows each molecular contribution to the fundamental bright collective phonon mode. Other parameters are specified in Supplementary Note S5.

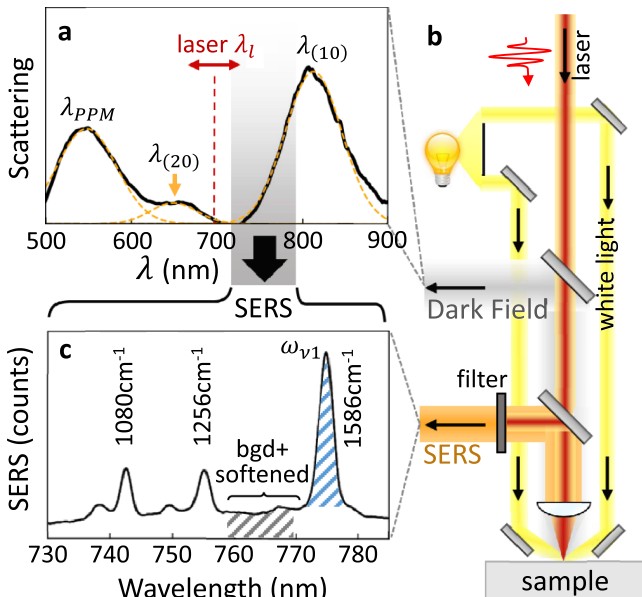

**Fig. 3 | Pulsed Raman scattering from plasmonic nanocavities. a** Dark-field spectrum of typical 80 nm nanoparticle-on-mirror (NPoM) containing BPT molecular SAM. Pump laser (red dashed) is spectrally tunable, shaded region shows range of SERS emission, with individual plasmon modes labelled. **b** Pulsed SERS experiment combined with white-light dark-field scattering on individual NPoMs. Spectrally-tuned 0.5 ps pump pulses excite individual NPoM (with white light off), and laser is filtered from the collected emission. **c** Pulsed Stokes SERS spectrum of BPT for three vibrational modes indicated. Blue shading shows 1586 cm$^{-1}$ peak area, grey shading shows region of softened mode + background (which is integrated for comparison).

larger than when only the molecule in the corner is present (corresponding to $N = 1$ in the figure) and 12 times larger than for a molecule in the gap centre.

We obtain the Raman spectra by applying the quantum regression theorem (Supplementary Note S1.2) to the patch of 100 molecules in the centre of the NPoM gap under continuous wave illumination. The obtained spectra (Fig. 2c) show a single narrow peak for weak illumination (blue line) at the vibrational energy $\omega_{v1}$ of the individual molecules, corresponding to the standard Raman line. In contrast, the collective SERS spectrum under strong pumping (red line) comprises a broad and strongly down-shifted line (by $\Delta_{os}^{N}$, arrow) associated with the fundamental collective bright mode (vertical dotted line), superposed on contributions from the remaining $N_m$-1 weak near-unshifted modes. The latter lead to a relatively narrow line close to $\omega_{v1}$ (blue line). Thus, the model predicts a redistribution of the scattered Raman signal from a frequency near $\omega_{v1}$, towards the softened frequency of the broad bright molecular phonon mode. The individual molecular contributions to the broad bright phonon mode are displayed in the inset of Fig. 2c with a clear symmetric maximum at the central position of the gap.

## SERS saturation effect in plasmonic nanocavities

Most SERS experiments use continuous wave (CW) excitation, where the spring shifts remain small (as discussed further below). In order to probe these optical spring shifts in a nanocavity, high instantaneous powers are demanded, which are first here supplied through pulsed excitation, together with NPoM constructs that localize light fields to a very small volume (thus increasing the local fields and the Green's function). The use of laser pulses together with these inhomogeneous fields however smears out the power-dependent SERS spectrum predicted by our model (Fig. 2c), although the effects appear consistent with what is observed as a repeatable

intensity-induced saturation of the sharp vibrational peak, as described below.

To explore the theoretical predictions, we use 0.5 ps laser pulses and concentrate on a strong SERS peak together with the region where the softened mode is expected to appear. For each of hundreds of NPoMs (depicted in Fig. 1a, see Methods for sample preparation), the plasmonic dark-field scattering spectra $DF(\lambda)$ and power-dependent SERS response $S(\omega_\nu, I_l)$ are characterized (Fig. 3). The plasmonic gap size, nanoparticle diameter, facet size, and gap contents control the spectral position of the main coupled plasmon $\lambda_{(10)} \sim 800$ nm. For the SAM of biphenyl-4-thiol (BPT) initially used, we concentrate on the strong $\omega_{v1} = 1586$ cm$^{-1}$ ring breathing mode producing Stokes emission from the pump pulse $\lambda_l$ at $\lambda_S^{-1} = \lambda_l^{-1} - \omega_{v1}/2\pi c$. Power series are recorded when blue-detuned from $\lambda_{(10)}$ ($\lambda_l = 633$ nm, $\lambda_S = 704$ nm), and compared with near-resonance emission ($\lambda_l = 700$ nm, $\lambda_S = 787$ nm $\sim \lambda_{(10)}$).

Pulsed laser excitation easily causes permanent damage from the high peak fields through irreversible chemistry or gold surface melting[14,25–27], so we develop here a strategy that employs the shortest possible illumination times, and measure many plasmonic nanocavities through fully automated experiments (see Methods). Pulses of duration 0.5 ps at 80 MHz repetition rate and average power $\leq 60$ μW are focused with a x100 dark-field high-NA = 0.9 objective to a sub-micron focus[14]. Dark-field images guide particle tracking software to concentrate only on well-formed NPoMs, monitor damage in real time, account for spectral aberrations, and minimize spatial drift. To reduce structural damage[14], the exposure time is scaled to keep the measurement fluence constant (10 μJ) while the average laser power ramps from 100 nW to 60 μW. To compare each NPoM $i$ with slightly different size and shape (limited by the precision of nanoscale fabrication) which varies their excitation and collection efficiencies, we normalize the results based on the integrated SERS counts at the lowest intensity $I_0$, using $\eta^i = S^i(\omega_{v1}, I_0)/I_0$. Given average $\bar{\eta} = \text{mean}(\eta^i)$, we then normalize the in-coupled intensity for each NPoM as $I_{in}^i = I_l \cdot \eta^i/\bar{\eta}$. This accounts for in-coupling efficiency so that each NPoM gives the same normalized SERS emission at the same low input intensity (see Supplementary Fig. S19). Enabled by this experimental correction, we explicitly present here the entire dataset on hundreds of NPoMs to show the reproducibility of the effect across many nanostructures and avoid bias incurred when selecting individual particles. However, all observations can be confirmed by data on individual cavities as presented in Supplementary Note S11.

Despite the intensity averaging from using laser pulses (which smears out the spectral shifts and makes direct identification of the $\Delta_{os}^{N}$ shift challenging), the evolution of the average SERS spectra for increasing $I_{in}$ shows clear nonlinear changes (Fig. 4a-c, normalized by in-coupled intensity). A repeatable weakening of the original sharp vibrational peak is seen in the power-dependent SERS, while the region at lower wavenumber grows superlinearly, indicating the energy redistribution into collective modes as predicted by the optomechanical theory (Fig. 2c).

To quantitatively analyze this SERS saturation, we extract the integrated SERS areas $S^i$ from the peaked emission around the $\omega_{v1} = 1586$ cm$^{-1}$ vibrational frequency (Fig. 3c, blue shaded). This is compared to the integrated emission from the softened peak region at lower wavenumbers (Fig. 3c, grey shaded), which however also contains a significant contribution from the SERS background. This ever-present background mainly arises from electronic Raman scattering (ERS) inside the Au[16]. At low powers ERS dominates this background, while for intense pulsed illumination it also contains the (smeared) softened modes from the broad collective phonon.

The integrated Raman peak emission $S^i$ shows a clear saturation with laser intensity, consistently for different vibrational peaks, NPoMs, pump wavelengths, and power series (Fig. 4d-f). To better show this, we also plot the SERS $\tilde{S}^i$ normalized to $\tilde{I}_{in}^i$ (Fig. 4g-i). The

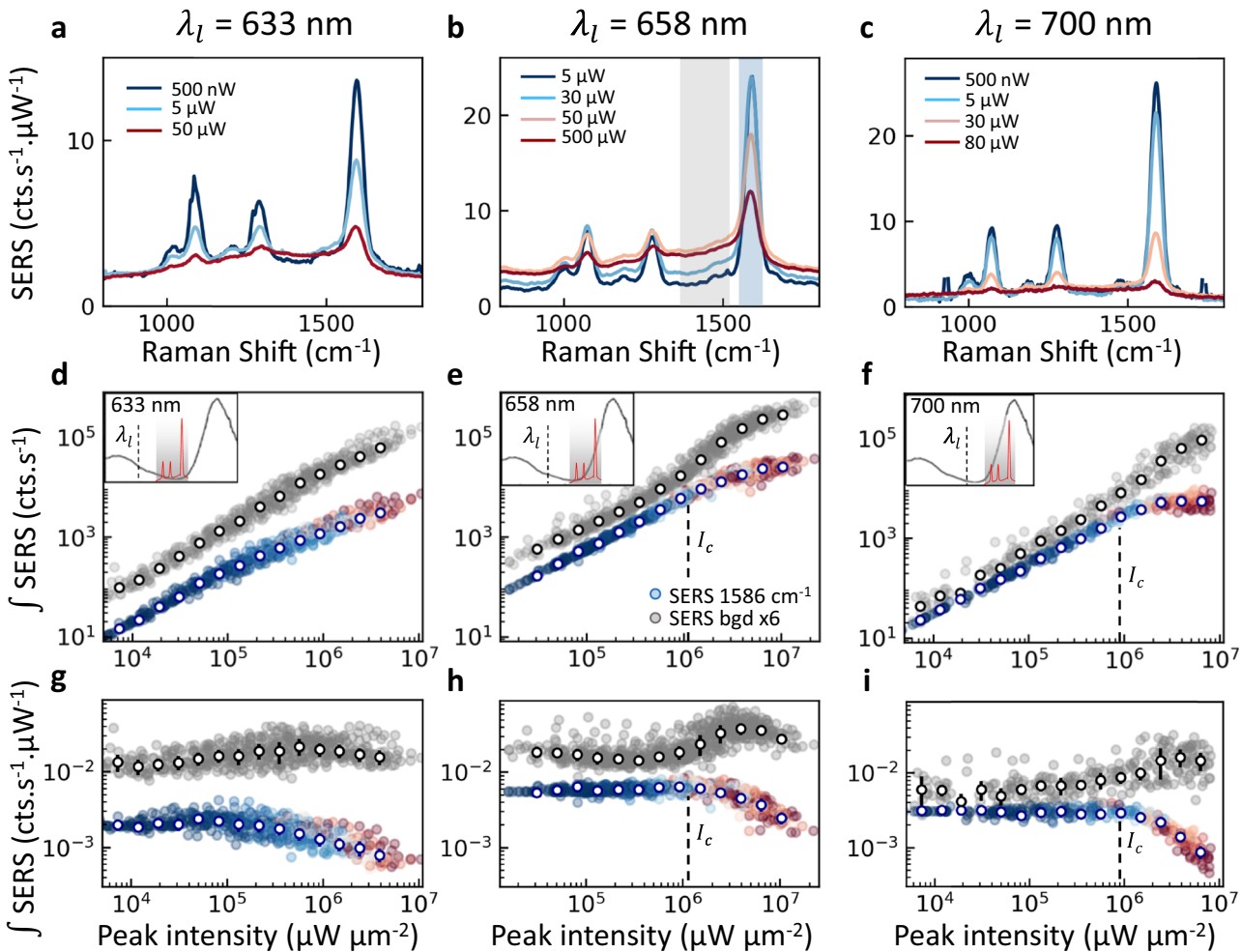

**Fig. 4 | Saturation of pulsed Raman scattering from many NPoMs. a–c** Averaged power-normalized SERS spectra for increasing in-coupled average powers at different pump wavelengths $\lambda_l$. **d–f** Integrated SERS emission from $\omega_{\nu 1} = 1586$ cm$^{-1}$ mode (colours show laser power) and integrated background + softened spectral region (grey, x6), excited by pulsed laser at $\lambda_l = 633$, 658, 700 nm. Open points are averages of individual measurements in each power range. Insets show relative position of pump wavelength and plasmon resonances. **g–i** Integrated SERS normalized by in-coupled peak intensity and integration time, error bars indicate their standard deviation. The critical laser intensity for saturation is marked as $I_c$ (indicated in the main text).

nonlinearity in pulsed SERS is dramatic, with up to ten-fold suppression of linear scaling of the sharp vibrational peak area at the highest pulsed powers. To confirm the saturation behaviour and eliminate damage or drift as the cause, power-dependent SERS measurements are repeated twice on each NPoM (Supplementary Fig. S23), showing that the saturation is reversible. Further power-cycling on each NPoM demonstrates that even for large SERS saturations, accompanying permanent damage of only 10% is seen. These observations contrast with literature reports on nonlinear SERS arising from irreversible damage under longer irradiation[28–30]. Vibrational pumping is also clearly observed in our data from the quadratic power dependence of the anti-Stokes emission (Supplementary Fig. S24), which then prevents meaningful temperature extraction (Supplementary Fig. S25).

Examining the high power spectra shows that as the vibrational peaks saturate, SERS emission in the softened mode region at lower wavenumbers correspondingly increases (see Fig. 5a, also evident in Fig. 4e, h with spectral integration range indicated in grey in Fig. 4b). This indeed points towards a redistribution of vibrational frequencies of many hundreds of cm$^{-1}$ (as predicted in Fig. 2c), consistent with unprecedentedly large softening of the vibrational frequencies in the optical nanocavity. This effect is very different from dc bias-induced vibrational Stark shifts where sharp lines shift by $\lesssim 10$ cm$^{-1}$ [31].

Additional experiments show that this SERS saturation is not specific to BPT, but is also seen for other molecules including naphthalene-thiol, triphenyl-thiol, and 4-mercaptobenzonitrile (Supplementary Figs. S26–S28), with similar saturation thresholds that depend on detuning (none have electronic transitions in the visible or near-infrared). We also find that molecule-metal hybridization is not a dominant influence, by substituting the first monolayer of atoms on the underlying Au mirror with Pd atoms. This changes the thiol binding but gives near-identical SERS saturation of BPT (Supplementary Fig. S29), showing that there is no power-dependent hybridization or charge transfer of the molecular orbitals. Similar behaviour is seen for pump wavelengths of 785 nm (Supplementary Note S13). In the minimal-dose regime shown here, previously observed superlinear SERS increases at high power[14] are still seen for some detunings (small initial rise in Fig. 4g) but at the higher powers that can now be achieved before bond-breaking, the SERS saturation is much more significant.

One other possible origin for these observations could be through the anharmonicity of the vibrational potentials, usually revealed through their thermal occupation. Transiently exciting molecules in solution[32–34] can give slightly red-shifted Raman peaks[35–38] arising from vibrational anharmonicity. However in our case, based on the reported biphenyl peak shifts with temperature[34,39,40], shifts of only $\Delta\omega_\nu < 30$ cm$^{-1}$ would result, much smaller than observed here

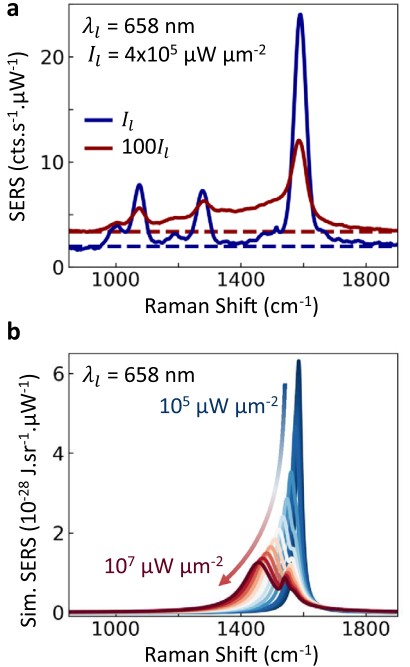

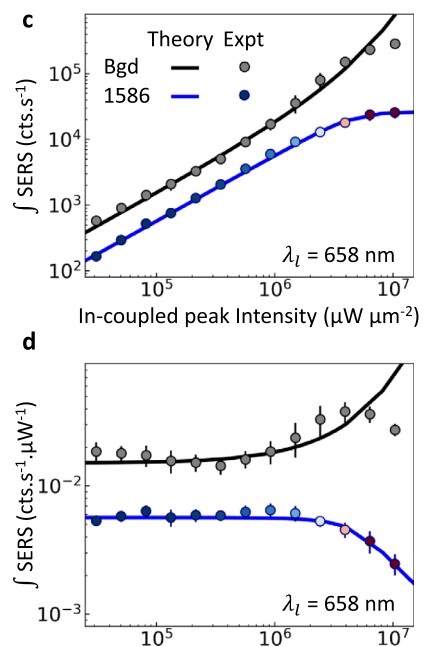

**Fig. 5 | Nonlinear vibrational coupling model vs expt. a** Experimental power-normalized SERS spectra at low (blue) and high (red) powers. Constant ERS background is estimated by dashed lines. **b** Corresponding theoretical results showing the SERS spectra vs CW illumination power, for 100 molecules arrayed around the gap centre. **c, d** Extracted Raman integrated in the region around the 1586 cm⁻¹ peak (blue line/symbols) and in the softened+background region between 1350–1500 cm⁻¹ (grey line/symbols) for theory and experiment (SERS normalized by power in **d**, as in Fig.4g–i). Experimental data are averages of many particles with error bars indicating standard deviation of individual measurements. In **c, d**, scaling of in-coupled power from theory by 0.24 is used to match with experiment. The transfer in weight from the 1586 cm⁻¹ peak to lower wavenumbers arises from the redistribution of emission to the red-shifted lowest-energy bright Raman collective mode.

(estimates in Supplementary Note S9). A model including anharmonicity cannot reproduce our results (Supplementary Figs. S15–18), instead suggesting that, even if contributing, a different interaction must be present with a much larger energy scale.

The correspondence between the spectral shifts predicted in theory and the experimentally identified saturation is analyzed in Fig. 5. The multimode optomechanical model indicates that above a critical pump power, a bright collective vibrational mode rapidly broadens and red-shifts linearly with power (Fig. 5b), producing a redistribution of vibrational frequencies that appears as a saturation of the originally-sharp SERS $\nu_1$ line and a superlinear increase in the softened + background region (Fig. 5c). We note that the saturation obtained with this model agrees well with the experiment (Fig. 5c, d), although the ERS and smearing mask the direct identification of shifts in Fig. 5a. In both theory and experiment, the total Raman yield integrated over all wavenumbers remains linear with power (Supplementary Figs. S8c and S24d, respectively), but is redistributed by the optomechanical interaction. To match the experimental and theoretical saturation it is only necessary to consider a slight scaling of the pump intensity. This scaling is not surprising since the calculations do not include all molecules in the gap due to the high computational cost (with different patches of molecules experiencing different intensity-dependent optomechanical coupling, Supplementary Fig. S9). Moreover, the exact spectral distribution of the collective Raman peaks and their relative weights will be influenced by the plasmon-mediated interaction of induced Raman dipoles (as noted above), as well as the specific configuration and orientation of molecules within the cavity. We note that although non-circular facets under the NPoM will shift plasmonic mode frequencies, the overall model still works in the same way.

We also can obtain a simplified analytical equation (see Supplementary Note S7) that predicts the critical illumination power needed to shift the Raman line outside its $\gamma_\nu = 20$ cm⁻¹ low-power linewidth (where saturation starts),

$$I_c\left[10^6\mu W \mu m^{-2}\right] \simeq a \frac{(2\pi c)^3 m_u \varepsilon_g d^3}{\eta_1' N_m EF^2 R_\nu^2} e^{(1.1\rho/d)^2}\left[Re\left(\frac{\varepsilon_{Au}-\varepsilon_g}{\varepsilon_{Au}+\varepsilon_g}\right)\right]^{-1}\omega_\nu[cm^{-1}]\gamma_\nu[cm^{-1}]$$

(4)

with proportionality constant $a = 1.2 \times 10^6$ (Supplementary Note S7), gap permittivity $\varepsilon_g = 2.1$, gap size $d$ and intermolecular spacing $\rho$ in nm, permittivity of Au at 658 nm $\varepsilon_{Au} = -13.5$[41], and $R_\nu \sim 960$ (in units of $\varepsilon_0 Å^2/\sqrt{amu}$, where $R_\nu^2$ is the Raman activity of the 1586 cm⁻¹ line, and amu $m_u = 1.7 \times 10^{-21}$ kg). The effective coupling to the collective Raman bright mode is accounted for by the factor $\eta_1(\rho) \approx \eta_1' \exp\left\{-(1.1\rho/d)^2\right\}$, here obtained as $\eta_1 \sim 0.12$ (at $\rho = 0.6$ nm) from the discussion in Fig. 2b. We note this simplification provides a useful intuition and compares well with the rigorous result used for Fig. 5, while omitting the dependence on plasmonic resonances. With the field enhancement factor $EF(\lambda_l) \sim 300$ of NPoM at the laser wavelength, for the 1586 cm⁻¹ mode, Eq. (4) gives $I_c \sim 3 \times 10^6$ µW µm⁻² at 670 nm for $N_m \sim 100$ molecules, in good agreement with the experiments. This formula shows why nanocavities are essential to bring the optical powers into a viable domain that does not damage the sample, since $I_c$ scales cubically with gap size and inversely with the power enhancement $EF^2$ (overall factors in excess of $10^6$ compared to free space). Equation (4) also clearly shows close-packed molecules in SAMs ($\rho < d$) are needed to observe such collective effects (though molecular ordering is not required).

The fractional reduction in bond strength from the optomechanical interaction by light is then

$$\frac{\Delta\omega_\nu}{\omega_\nu} = b\eta_1 N_m\left(\frac{EF\,R_\nu}{\omega_\nu}\right)^2 I_l,$$

(5)

where for $\omega_\nu$ in cm⁻¹ and $I_l$ in $10^6\mu W\mu m^{-2}$, $b = 2.23 \times 10^{-8}$. The relative lineshift can become even larger for low frequency lines, implying that

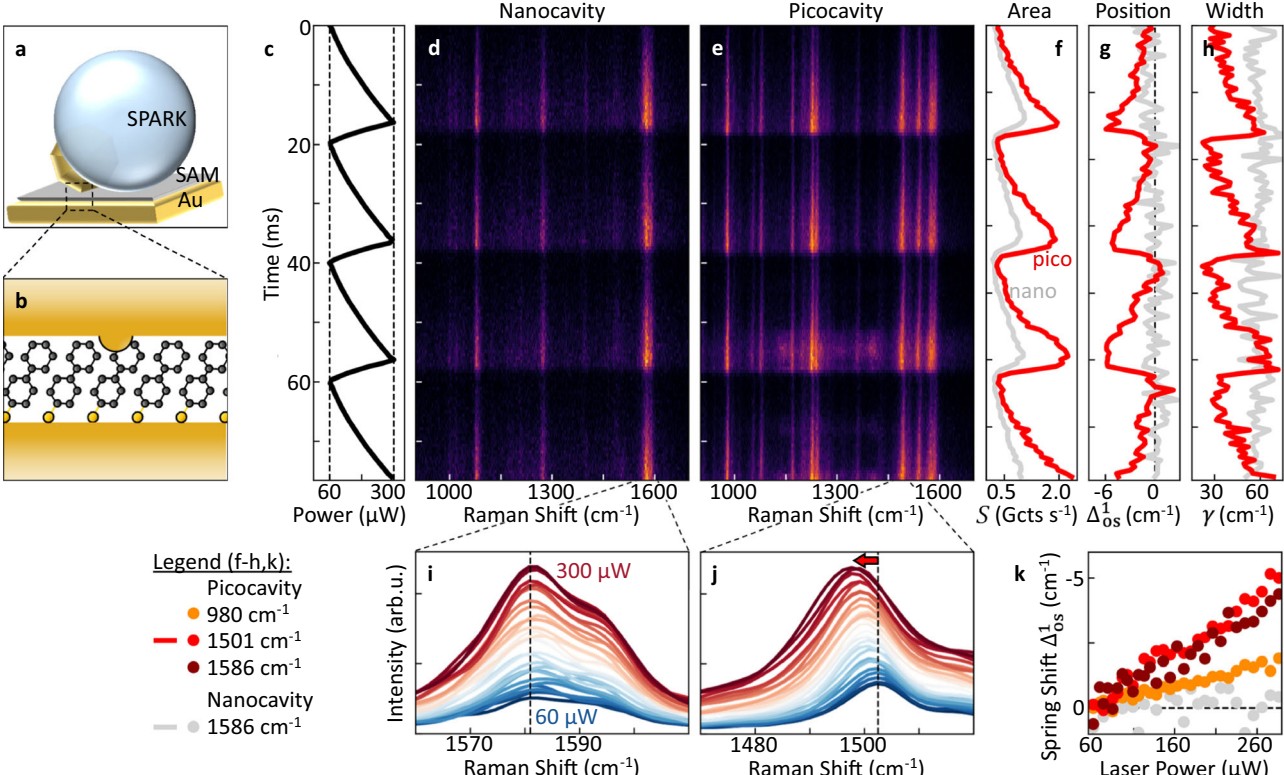

**Fig. 6 | Optical spring shift in picocavities. a** Schematic of nanolens on NPoM (SPARK construct). **b** Generation of a picocavity when Au atom moves onto facet, enhancing field at a single BPT molecule. **c** Sawtooth modulation of 633 nm CW laser power from 60 to 300 μW at 50 Hz. **d, e** Fast spectral scans (0.5 ms integration time) of Stokes emission from the SPARK nanocavity (**d**), and after formation of a picocavity (**e**). **f-h,** Extracted fits to 1586 cm⁻¹ line in nanocavity (grey) and 1501 cm⁻¹ picocavity line (red). Peak area (**f**) is linear in laser power, while optical spring effect in the picocavity leads to a repeatable shift in position (**g**) and broadening (**h**) of the vibrational line. **i, j** Spectra of vibrational lines investigated in f-h averaged over 4 periods of laser modulation. Nanocavity line (**i**) shows constant width and position while picocavity line (**j**) shifts and broadens (colour gives laser power). **k** Optical spring shift $\Delta_{os}^1$ dependence on laser intensity for several vibrational lines in nano- and picocavities. Each picocavity vibration experiences a different optical spring magnitude.

irradiation even more strongly weakens rotations, librational and shearing deformations of molecules and macromolecules. As a result, for instance, enzymes might be optically-dressed and switched through this scheme of interactions (note proteins and lipids have already been placed in such plasmonic nanocavities[42]).

## Single-molecule optical spring shift in picocavities

Whilst pulsed excitation of nanocavities red-shifts and smears out SERS spectra (as Fig. 5a), direct visualization of the optical spring shifts is desirable. This requires CW illumination, but to reach observable spring shifts requires extra field enhancement. To show this we use NPs with integrated nanolenses (called SPARKs to enhance coupling efficiencies[43], Fig. 6a) which support picocavities inside them[44] (formed when single gold adatoms are pulled from the facet into the gap) that additionally confine light below the 1 nm scale[12,45] (Fig. 6b). Picocavities, as compared to nanocavities (which lack the atomic protrusion), give particularly strong optomechanical coupling and focus the optical fields down to the scale of an individual molecule[12,45].

As previous work on picocavities has shown, SERS lines in picocavities fluctuate in position on multi-second timescales due to the single-molecule nature of the signal[12,45,46]. Hence, the meaningful extraction of the optical spring shift is experimentally challenging. Here, we greatly increase the speed at which power sweeps are performed, by 10⁴ compared to previous investigations[12], to avoid slower spectral wandering. A 633 nm, 300 μW CW laser is used to acquire Stokes spectra at kHz rates (Fig. 6c–e, see Methods). A sawtooth modulation of laser power from 60 to 300 μW at 50 Hz (Fig. 6c) repetitively probes the power dependence of the SERS in the

nanocavity and picocavity within 20 ms, faster than any spectral drift (Fig. 6d, e, respectively). Fitting the vibrational lines with a Lorentzian peak gives the area, position, and linewidth *vs* laser power (Fig. 6f–h). For the picocavity lines, increasing illumination leads to a reversible red-shift in the vibrational energy as well as strong broadening by a factor of 2, while vibrations in the nanocavity remain unchanged. This behaviour can be clearly seen in power-dependent spectra averaged over four laser modulation cycles for the nano- and picocavity (Fig. 6i, j respectively). Extracting the position of different vibrational lines shows that all vibrational modes in the picocavity experience a different shift rate (Fig. 6k), but are always reversible. The strongest shifts of 5 cm⁻¹ are observed for the 1501 cm⁻¹ and 1586 cm⁻¹ lines in the picocavity (Fig. 6k).

This observation of a reversible peak shift with laser power in picocavities is found to be widely reproducible across many particles and picocavity events (further data provided in Supplementary Fig. S30). A detectable spring shift was found in over two thirds of stable picocavity events recorded from 150 SPARKs, with undetectable shifts only for particles with weaker light in-coupling. A combined statistical analysis of all picocavities (as conducted above for nanocavities) is however not possible, since each picocavity is characterized by a different set of vibrational lines[45].

We now compare the observed shift for CW excitation of picocavities in SPARKs to the shifts in our pulsed experiments in NPoMs (Figs. 4, 5). Equation (5) shows that the spring shift depends on the local field and the number of molecules (including correction factor $\eta_1$). Using the experimentally observed ratio of SERS from SPARK nanocavities and standard NPoMs, we estimate the field in the latter is

7-fold smaller[43]. Additionally, near-field focusing around the atom tip of picocavities enhances the local field further by ~3 times (which is consistent with observed experimental enhancements of SERS from picocavities of $3^4$-100). Collective vibrations lead to a 12-times larger spring shift for 100 molecules than a single molecule (see Fig. 2b). The CW laser has a much lower power than the pulsed peak laser power, $I_l(\text{CW})/I_l(\text{pulsed}) = 3\times10^{-4}$. Assuming that the Raman cross-section of molecules in the picocavity does not change (full calculation not yet possible), the pulsed nanocavity containing several molecules should give 100 times larger shift than a single molecule in a CW-pumped SPARK picocavity. Experimentally, we observe a shift >250 cm$^{-1}$ with laser pulses compared to ~cm$^{-1}$ in SPARK picocavities, giving a factor of ~50 consistent with the above approximations. For comparison, in SPARK nanocavities (Fig. 6i) the spring shift is below the instrument resolution due to the ~3 times lower optical fields. Importantly, we note that the analytical model in Eq. (5) uses a simplified description of the plasmonic modes of the metal-dielectric nanostructure. Since the plasmonic modes of SPARKS are not yet known in detail (since dark-field scattering is obscured by reflections from the silica microlens), the above approximation can only confirm that the order of magnitude of the spectral shift observed is consistent with the optomechanical spring shift expected. Further, modifications of the Raman tensor induced by the picocavity might give different field enhancements and effective number of molecules involved in the picocavity-induced optical spring. However, the consistency of our estimates reinforces the optomechanical origin of the experimental results, while these picocavity observations directly evidence the repeatable spring shift from optomechanical coupling.

The observed vibrational shifts might also arise from vibrational anharmonicities under strong vibrational pumping in picocavities[12]. From our above model of the anharmonicity and linear scaling of this shift with laser intensity (Supplementary Note S9), we estimate the effect to be one order of magnitude smaller than the shift observed here. While not sufficient to fully discard contributions from vibrational anharmonicity, further detailed work on anharmonicity in picocavities is thus also needed.

## Discussion

The effect of the optomechanical interaction of the molecules, apparently observed here in plasmonic nanocavities, is to adiabatically decrease the bond strengths during a pulse (as the optical spring squeezes them). This effect is analogous to the Lamb shift in excitonic emitter-cavity coupling. It provides a new way to manipulate bonds which should be contrasted both with coherent control (based on electronic wavepacket excitation of light-absorbing molecules[47]), and with vacuum Rabi splitting (based on infrared light-matter coupling without any light present[48]). In the latter, strong coupling at mid-infrared frequencies causes energy to cycle between vibrational dipoles and photons, completely different from the optomechanical effect here where optical radiation pressure weakens the bonds themselves.

Our results indicate that in nanocavities, vibrational shifts much larger than the linewidths (≫20 cm$^{-1}$) can be attained, comparable with the largest vibrational strong-coupling Rabi splitting[49]. The estimated $\Delta\omega_\nu$ implies fractional bond energy reductions of >10% (exceeding thermal energies and with minimal damage), corresponding to light-controlled weakening of the bond spring constants by $\sqrt{(\Delta\omega_\nu/\omega_\nu)} \sim 25\%$. Further pumping seems to reach light-induced dissolution of the collective bond, leading to irreversible bond breaking. Such effects are impossible to observe for molecules in solution[32–34] which are too far apart (≫$d$) to coherently couple, and without the nearby metal surface they show negligible shifts.

The shifts also depend on the number of molecules hybridizing to give bright coherently-coupled phonon states. We thus explored mixed SAMs with 50% TPT and 50% BPT (as well as other fractions), and

find that the power threshold is little affected (Supplementary Fig. S28). The similar vibrational spectra of these molecules thus suggests that hybridization also occurs between distinct but similar molecules.

We conclude by summarizing our results in the context of optomechanics. The large value of Re{$G$} associated with the short-ranged interaction dramatically enhances the optical spring effect in such plasmonic gaps. Indeed, calculating the optical spring effect for single molecules in our NPoM gap as a function of laser wavelength (Fig. 2a) shows the light-induced spectral shift of the Raman line of a molecule, $\Delta_{os}^1 \propto \text{Re}\{G\}$, can be more than hundred-fold enhanced over that obtained in traditional dielectric cavities. This is a consequence of (i) fully including image charges from the transient Raman dipoles through the Green's function (tenfold enhancement), and (ii) Re{$G$} being positive at both Stokes and anti-Stokes frequencies, so that the two contributions in $v_{ss} = (S_{ss}^+)^* + S_{ss}^-$ add up instead of largely cancelling, as occurring in single-mode optical resonances (another tenfold enhancement, Fig. 2a, Supplementary Note S6). We illustrate the difference in Fig. 7a by plotting the ratio of the spring shift to the vibrational linewidth $\Gamma_{\text{tot}}$. The latter includes the optomechanically-induced broadening (or narrowing) of the vibrational losses (optomechanical pumping), and is shown at laser intensities chosen so it equals half the vibrational losses, $|\gamma_\nu - \Gamma_{\text{tot}}(I_l)| = \gamma_\nu/2$, at the frequency that maximizes $\Gamma_{\text{tot}}(I_l)$. The colour map behind shows the results for single-mode cavities as a function of the cavity losses $\kappa_c$ and detuning of the incident laser. The corresponding ratio for the NPoM (shown in the box) is significantly larger than for a single-mode cavity of similar losses, and is less dependent on the frequency of the laser.

The net effect of the NPoM cavity according to Eq. (1) is for the Raman peaks to linearly shift and broaden with laser power, even for a single molecule. The resulting spring shifts per photon in the cavity are correspondingly higher than other systems[50] (Fig. 7b), for vibrations in the ground state at 300 K, and offer further enhancements accessible from improved nanocavities or collective effects (arrow), even potentially exceeding the linewidths of cavity and vibrations.

The continuum-field optomechanical model for multimode plasmonic cavities reveals that ultrasmall mode-volume plasmonic nanocavities yield SERS emission from pumped molecular collective vibrations red-shifted by 10 to 100-fold more than in single-mode systems, leading to a redistribution of energy and a saturation of emission from the original sharp vibrational lines. This physical picture is commensurate with experimental observations of SERS saturation under pulsed illumination, and reversible SERS lineshifts under CW illumination of picocavities. Independently, the two experiments cannot unambiguously confirm the existence of the proposed optical spring shift, however both are in good agreement with the optomechanical model developed here. This also gives implications beyond simply limiting the maximum Raman yield from molecules (to <10$^{12}$ counts.s$^{-1}$)[16]. In the plasmonic cavities, light transiently softens the bonds of molecules near metal interfaces, which may find use in optical catalysis of reactions and photodecomposition for recycling, as well as controlling molecular photodetectors and other molecular nanoscale optoelectronic devices. We also note that permanent damage to molecules (see Supplementary Note S12) occurs when bond softening starts, suggesting its mechanism may be related. Such plasmon-induced bond softening opens up fruitful possibilities to explore correlations of vibrations at room temperature, since $\hbar\Delta\omega_\nu > k_BT$. The convergence of molecular electronics, plasmonics, quantum emitters, and vibrational coherence gives opportunities for using quantum-correlated SERS to probe electronic transport, dissipation and switching. We emphasize that the results here operate not just for molecular layers, but also for 2D layered crystals such as transition metal dichalcogenides or graphene, and will lead to their drastically different optomechanical device operation when paired with plasmonic nanocavities.

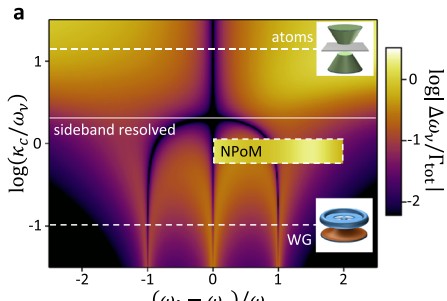

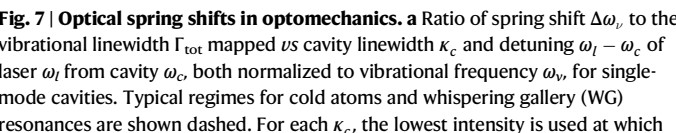

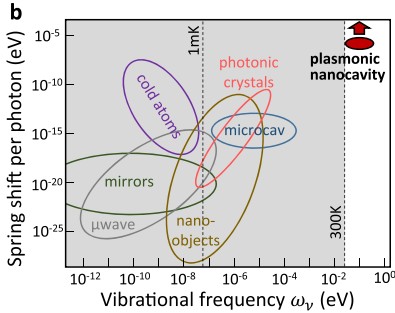

**Fig. 7 | Optical spring shifts in optomechanics. a** Ratio of spring shift $\Delta\omega_\nu$ to the vibrational linewidth $\Gamma_{tot}$ mapped *vs* cavity linewidth $\kappa_c$ and detuning $\omega_l - \omega_c$ of laser $\omega_l$ from cavity $\omega_c$, both normalized to vibrational frequency $\omega_\nu$, for single-mode cavities. Typical regimes for cold atoms and whispering gallery (WG) resonances are shown dashed. For each $\kappa_c$, the lowest intensity is used at which $|\gamma_\nu - \Gamma_{tot}(I_l)| = \gamma_\nu/2$ over the detuning range. Box shows the results for the NPoM including the full multimode plasmonic response, as considered in the simulations, and with $\omega_c$ fixed at the NPoM dark-field resonance of 800 nm, giving ~10-fold enhancement. **b** Comparison of spring shifts per cavity photon *vs* $\omega_\nu$ for a range of systems[50].

## Methods

### Optomechanical simulations
A detailed description of the optomechanical theory employed to calculate SERS spectra is provided in the Supplementary Information.

### Sample preparation
The Au substrates are prepared via a template stripping method, which has been detailed elsewhere[51]. Briefly, atomically flat Au surfaces are produced by evaporating 100 nm of Au onto Si wafers at a rate of 0.5 Å/s. Small pieces of silicon are then glued to the wafer using epoxy, and the wafer slowly cooled from the 150 °C curing temperature to room temperature. These silicon pieces can then be peeled off to reveal a smooth Au surface with rms roughness <0.2 nm. The SAM is prepared on the Au surface by immersion in a 1 mM analyte solution in anhydrous ethanol (>99.5%) for 22 h. The nanoparticles are purchased in suspension from BBI Solutions (80 nm, OD1, citrate capped). They are dropcast onto the SAM for 30 s before being rinsed with deionised water. The short time used for dropcasting ensures a low density across the sample so they can be individually observed in optical microscopy. Aggregation is prevented by citrate capping around the Au NPs.

### Pulsed SERS spectroscopy on NPoMs
In a custom-built, inverted darkfield microscope NPoMs are located automatically by particle tracking algorithms and centred by moving the sample stage. On each NPoM structure, Raman and dark-field spectra are acquired in quick succession. For pulsed Raman spectroscopy, a Spectra-Physics Maitai laser at 80 MHz is used to drive an optical parametric oscillator (Spectra-Physics Inspire), producing 100 fs pulses of tunable wavelength. For the required spectral resolution, pulses are filtered by a tunable bandpass filter (PhotonETC LLTF contrast) to 1.5 nm spectral bandwidth and 500 fs pulse duration. The light scattered by the sample is filtered by a Fianium Superchrome tunable longpass filter and detected by a Raman spectrometer. For power-dependent experiments, average laser power is ramped from 100 nW to 60 μW while adjusting integration times to keep constant fluence (100 s integration time for 100 nW, decreasing to 167 ms at 60 μW). We follow thousands of SERS spectra by examining hundreds of individual NPoMs for long periods of time. Correction for NPoM in-coupling efficiency is described in Supplementary Note S10.

The integrated SERS emission of the Raman lines is obtained by calculating the area underneath the spectra in a 40 cm⁻¹ wide window centred on the peak. The SERS background area is obtained by integrating a 150 cm⁻¹ wide window between the Raman lines (1350 cm⁻¹ to 1500 cm⁻¹). To compare the acquired spectra with theoretical calculations, the average in-coupled laser power is converted to the in-coupled peak power of the pulsed laser. For our laser pulses of 0.5 ps duration and 80 MHz repetition rate, an average power of 1 μW corresponds to a peak power of $2.5 \times 10^4$ μW and a peak intensity of $3.2 \times 10^4$ μW μm⁻².

### Fast Raman power sweeps on SPARKs
Fast Stokes spectra acquisition is carried out on Au nanoparticles with an integrated silica nanolens, termed "Superefficient plasmonic nanoarchitectures for Raman kinetics" (SPARK). SPARK constructs are produced by organosilica synthesis using Au NPs as seeds for nucleation and growth[43]. The SPARK particles are then deposited according to NPoM sample preparation described above. Such SPARK samples provide sufficient signal to allow the acquisition of SERS spectra with sub-ms integration times. To repetitively probe the power dependence of SERS spectra, a 633 nm continuous-wave laser is modulated with an acousto-optic modulator driven by a sawtooth voltage at 50 Hz supplied by a function generator. The power modulation is calibrated by measuring the minimum and maximum power with a power meter and characterising the shape of the modulated SERS background. Spectra are acquired with an Andor Newton 970BVF using the Fast Kinetic readout mode. An automated darkfield microscope is used to scan hundreds of nanoparticles and collect successive kinetic spectral scans for several minutes on each particle. The spectra are then screened for the generation of pico-cavities (characterised by the emergence of new intense Raman lines) and the optical spring shift is analysed by fitting Lorentzian peaks to each individual vibrational line.

## Data availability
The figure data in this study are deposited in the Cambridge open data archive under https://doi.org/10.17863/CAM.95259.

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

## Acknowledgements

We thank Mikolaj Schmidt and Adrián Juan Delgado for fruitful discussions, and Marlous Kamp for synthesizing SPARK constructs. We acknowledge EPSRC grants EP/N016920/1, EP/L027151/1, NanoDTC EP/L015978/1, NSFC grant 12004344, NSFC-DPG grant 21961132023, Basque Government grant IT1526-22, grant PID2019-107432GB-I00 funded by MCIN/AEI/10.13039/501100011033/, and EU THOR 829067, POSEIDON 861950 and PICOFORCE 883703. L.A.J. acknowledges support from the Cambridge Trust and EPSRC award 2275079. B.d.N acknowledges support from the Winton Programme for the Physics of Sustainability, and the Royal Society in the form of a University Research Fellowship URF \R1\211162. C.C. thanks NPL for PhD funding.

## Author contributions

W.M.D., E.P., L.A.J., B.d.N., S.H., C.C. and J.J.B. devised the experimental techniques and developed sample fabrication; W.M.D., E.P., L.A.J., J.J.B. developed the spectral analysis; Y.Z., J.A., R.E., T.N., J.J.B. developed the simulations and models; all authors contributed to writing the manuscript.

## Competing interests

The authors declare no competing interests.
