## [Peer review file · Nature Communications]

Reviewer comments

Reviewer #1:

The authors have provided a detailed and convincing response to most of my comments and have modified the manuscript accordingly. The present version conveys more clearly the message and relevance of their findings and differentiate from other works (it was already clear to me that the off-resonant optomechanical coupling control was the key result, however my main concerns were not with the experiments but with the underlying theoretical modeling, that spite of qualitatively reproducing the experimental observation still is not satisfactory as it does not provide physical insight into the microscopic mechanism behind the shift for just one mode). Since the first round I found the present experimental observation remarkable and highly interesting and novel (I apology for the error in my previous report, it should have read novel and unexpected). I fully agree that they clearly show a optomechanical spring effect of a molecular bond mediated by the nanocavity. In that respect I think the experimental findings of this work make it suitable for publication in nature nanotechnology.

> We thank the reviewer for their positive appreciation or our work, which they find suitable for publication. Indeed as suggested we now clarified the microscopic mechanism in the main text before discussing the experimental results.

Reviewer #2:

I was positive about publication in my original review, and I remain positive. The more critical first reviewer was confused about the phenomena being studied, as should be clear from the authors response. The authors have now enhanced their development in the SI so as to explain the distinction between these different kinds of experiments, which is a welcome addition. I was also favorably impressed by the authors response to my (less serious) criticisms, and I have nothing new to add. The only unfortunate aspect of the manuscript is that the beautiful (and important) theoretical development is "buried" in the SI.

> We thank the reviewer for their positive appreciation of our work, which they find suitable for publication. We took their helpful advice into account, and have completely rewritten the paper to first present the important theory aspects, and then connect the predictions to our specific experimental findings.

Reviewer #4:

> While reviewers 1 and 2 were happy to see this published in Nature Nanotechnology, reviewer 4 requested more clarity. We thus now moved the manuscript to Nature Communications, which enabled us to completely rewrite the paper leading with the theoretical background. This version reveals more clearly the physics behind the effects and how the theoretical prediction is supported by the experiments.

The paper claims an optical field induced softening of molecular vibrations. However, in reviewing the manuscript and the other review reports, it is hard to see

1. Whether there is really a frequency shift in the data;

> It is not possible to directly see a simple shift in experiments, due to (a) the ever-present strong SERS background in this region, (b) the need to use pulses to access the powers required which broadens all SERS peaks, and (c) the smearing of the shift that each pulse produces from its rising/falling power in the inhomogeneous local fields. We thus study a different signature of the optical spring effect consisting of the redistribution of energy between the narrower unshifted peak and the broader strongly shifted new peak. Due to this redistribution, photons emitted around the

narrow peak saturate while the light emitted in the region of the broad peak increases nonlinearly. This effect is clearly observed experimentally by integrating the spectra over different spectral ranges and is in good agreement with the theoretical prediction.

To avoid confusion, we now write explicitly *“The use of these laser pulses... smears out the power-dependent SERS spectrum predicted by our model as a direct spectral shift (Fig. 2c), however the effects are observed as repeatable intensity-induced saturation of the sharp vibrational peak, as described below”, “Despite the smearing from using laser pulses (which makes direct identification of the shift challenging), the evolution of the average SERS spectra for increasing I_{in} shows clear nonlinear changes (Figs. 4a-c, normalized by in-coupled intensity)” and “The correspondence between the spectral shifts predicted in theory and the experimentally identified saturation are compared in Fig.5.”*

2. Whether if so, it could not be explained by structural, orientational, DC or optical Stark shifts, or other effects;

> We explicitly consider in the SI the two main effects (heating and anharmonicity) that in our view could interfere with the effects observed here, and give strong reasons why we believe they can be discarded here. Vibrational Stark shifts are much smaller [see Nature Catalysis 4, 157 (2021)] for any reasonable voltages. The dynamical shift of emission is precisely addressed through the Lamb shift. While not possible to discard structural or orientational changes of the molecules, we see no compelling reason to expect that it would result in the effects seen. Further, our trends are reversible (Section 12 of the SI), while one expects such molecular changes to be irreversible.

3. What evidence there would really be that is it associated with an opto-mechanical effect;

> Firstly, we prove that we are in the optomechanical regime from the quadratic power dependence of antiStokes emission. We believe optomechanical interactions drive the experimentally identified effects as only this agrees with our theoretical predictions (see below). In the new version we are careful not to claim direct observation of a simple vibrational shift. For example, we write now in the abstract *“The theoretical results are consistent with the strongly non-linear behavior exhibited by the Raman spectra of molecular monolayers...”*, and in the introduction *“Here for the first time, we report indications of a vibrational frequency shift associated with an optical spring effect”* and in the conclusion *“This physical picture is validated by experimental observations of SERS saturation under pulsed illumination”*. As do all the other reviewers, we believe that the theoretical prediction of this unexpected effect is already an important result in itself, which is strongly reinforced by the good agreement with the experiment.

4. And/or why it would be connected to the (in fact quite different) predictions of ref. 10 and 11, which are in themselves controversial claims (and for which as of today no experimental results exist that would support the theoretical claims).

> We thank the reviewer for their comments, which motivated us to completely rewrite the paper. In our new version, we first present the theoretical model and the physical phenomena that are expected from this model. We then describe the experiments and their connection with the theory. We note that experimental support of [10,11] is already found in [12,14], with detailed discussions in [13], as well as new works: PRX 10, 031057 (2020), arXiv:2107.02507, and arXiv:2107.03033.

Our results connect to [10] because the same theoretical description of SERS based on optomechanics is used. The effect studied is however very different. For the system considered in [10,11] (single plasmonic mode and single or identical molecules), the shift is much smaller and there are no collective modes, or saturation effect.

5. In agreement with the reviewer 1 (items 1 and 2), and not addressed by the response from the authors, this reviewer equally raises concerns about a theoretical construct that is mapped onto the data yet lacks a clear physical interpretation of what should give rise to that vibrational softening.

> The vibrational softening is a direct consequence of the optical spring effect, which is a phenomenon well-known in the optomechanics literature [see for example Rev.Mod.Phys. 86, 1391 (2014) and references therein]. This spring effect is greatly enhanced in our nanocavity (compared to the single-mode single-molecule case) because of:

a) the formation of collective phonon modes due to molecule-molecule interactions (mediated by their coupling with the plasmons). To better illustrate this point, we now include in the main text a new Fig. 2b (previously placed in the SI). This figure shows the enhancement of the spring effect with an increasing number of molecules.

b) the excitation of mirror charges in the metallic surface. The effect of the mirror charges can be large because the distance between the molecule and the metal surface is very small. These mirror charges are not included in a single-mode model (the standard description in optomechanics) and strongly enhance both the molecule self-interaction and molecule-molecule interactions that cause the spring shift effect. The large enhancement of the spring shift in our cavity for a single molecule, compared to the result of the single mode, is shown in Fig.2a.

We believe that this physical interpretation is much clearer in the new version of the manuscript. We also emphasize that these effects are contained in the full Hamiltonian discussed in the SI, which is a direct generalization of the single-mode single-plasmon model to our situation.

6. The paper lives more on trying to assure the reader it were so, rather than providing data linked scientific evidence. It also suffers under embellished and scientifically imprecise language of ‘optical spring’, ‘optomechanical dressing’, etc.

> We believe these are all widely-used terms in the optomechanics community. For instance “Optical spring effect” is not imprecise language but carefully defined [see for example Rev.Mod.Phys. 86, 1391, (2014) which is one of the key references of the field]. As discussed generally [e.g. in Physics 2, 40 (2009)], it is connected with the gradient of the radiation force induced at the position of the molecule. The modification of excited states by strong laser illumination (for example to create the Mollow triplet) is conventionally referred as ‘dressing’ (or ‘dressed states’) in quantum optics. Thus, using ‘optomechanical dressing’ to describe the changes of vibrational energies induced by the optomechanical interaction with sufficiently strong lasers is a direct generalization of this terminology.

7. Overall, the manuscript confuses rather than convinces a scientific reader. The paper is written in a very difficult way to even get to the bottom of it.

> In response to this, we decided to completely rewrite the paper and expand the theoretical presentation so that it becomes much clearer, and reverse the order of theory and experimental evidence, as to provide a better flow of the concepts.

8. Just several specific points about the data, and the derived claim of a frequency shift, not even touching on its interpretation: One big concern and center point of the claim is that there doesn't seem to be enough evidence to prove that the authors measured a shift in the Raman peak.

> As discussed in detail above, the shift is somewhat smeared out, but can be identified through the redistribution of Raman photons emitted at different energies. More direct observation of simple shifts is not currently feasible using pulsed light on realistic plasmonic nanocavities, which are the required conditions to access the regime of this optomechanical effect.

9. In Fig. 2b and Fig. 4a, what evidence is there that the 1586 wavenumber peak has shifted to the background region? The background has increased, but that could just be due to the normalization as the system saturates. It would be nice to see a Lorentzian fit added to each of the Raman peaks and also to the background at 1450 wavenumber. If the background really increased due to bond softening and a redshift of the Raman mode then one would expect that background area to have a Lorentzian shape, but it doesn't look like that is the case. In Fig. 1b the dashed line shows that the shifted Raman peak should be a Lorentzian shape, but their experimental data in Fig. 2 shows a very flat background signal. If it is the shifted Raman peak then it has a huge linewidth and that should be discussed.

> As discussed above, it is not possible to directly observe a simple shift of the peak in experiment. We do not understand what the reviewer means by "could just be due to the normalization as the system saturates". Special care has been taken so that all normalization is done in a consistent manner (both when comparing between samples, and when the intensity dependence is analysed for each individual NPoM construct). A conventional Raman model predicts a linear increase of the background and a linear or superlinear increase (i.e. not a saturation) of the Raman line. To the best of our knowledge, and that of all other reviewers, no other effect could give rise to these observations. The comment "normalization as the system saturates" seems to us rather vague without presenting any alternative to our model. We emphasize again that the measured effect is predicted by a Hamiltonian that is a direct generalization of the models previously used in molecular optomechanics (we do not include any new effect 'by hand'), and is fully supported by the experimental evidence.

10. According to equation 2, v_{ss} is proportional to the laser power. According to eq 1, the vibrational frequency will change linearly with v_{ss} and thus linearly with the laser power. So one would expect this to cause the Raman peak to shift linearly as the laser power increase. This is what happens in the simulation in Fig. 4b but that is not what they measure experimentally. In Fig. 2a-c the intensity of the Raman peaks "leaks" into the background area creating more of a dual-peak structure instead of shifting the original peak gradually as the power increases.

> The signature of the optical spring effect that can be measured in the experiment is the saturation of the energy emitted from the Raman peak, and the superlinear increase of the background. We write explicitly in the new manuscript that the simple shift cannot be measured directly, but rather through this saturation of the signal. We believe that the good agreement shown in Figure 5c,d between the experimental and theoretical results (evolution with laser intensity of the background and of the signal near the 1586 cm^{-1} Raman peak) supports the validity of our model.

11. In lines 155-158 they claim that the 'optical spring effect' arising from the self-interaction of the Raman dipoles ($s=s'$) is what causes the shift in the Raman lines. Then the paragraph starting at line 189 says that the interaction between the Raman dipoles also causes the shift. This is confusing at best. Which or both effects are significant here? Does one of these effects dominate over the other? This leaves a confusing picture of what the authors actually attribute the shift to, and again what exactly in the data would solidify the claimed frequency shift.

> This is a helpful clarification. The frequency shift is explained by the optical spring effect, which is due to two closely related contributions: the self-interaction of a molecule with itself and the interaction between molecules. Both are mediated by the NPoM. Typical optomechanical systems only contain one vibrating element (e.g. one molecule) so this optical spring effect is associated with only the self-interaction. With many molecules present, the shift induced by the interaction between molecules can be understood as a generalization of the optical spring effect. As we describe now in the main text (Fig.2b) the contribution to the optical spring effect due to molecule-molecule

interactions becomes dominant for moderate and large number of molecules. We also note that, from a different perspective, one can understand this same effect as the optical spring effect experienced by the collective phonon modes that are induced by the optomechanical interaction.

To avoid this confusion, we now start the paragraph discussing the self-interaction with *“We focus first on the vibrational shift experienced by a single molecule at the center of the gap. The real part of the spectral density of the molecule experiencing the plasmonic NPoM, $Re\{S_{ss}^{\pm}\}$, associated with the Stokes (+), or anti-Stokes (-) frequencies respectively, determine the total frequency shift of each vibrational mode”*. The fact that the molecule-molecule interactions increase the spring effect is introduced explicitly by the following sentence: *“This significant optical spring effect in molecular self-assemblies is further collectively enhanced by coupling of molecules laterally... from their image-dipole local Coulomb interactions”*.

12. In Fig 4d and in Fig 2d-i there are larger uncertainty bars in the data points after the critical intensity (see the 3rd to last point in Fig 2i or the 5th to last point in Fig 4d as examples). These measurements are essential for determining the critical intensity when saturation begins. Without those points, the curves would look approximately linear. Since these points are so important it would be good to address why they have a large uncertainty.

> Again, this is a helpful point. The error bars shown in Fig 4d and Fig 2d-i show the standard deviation of individual nanoparticle measurements in a given power range. This variation of individual measurements will thus be higher in ranges with strongly varying signal (i.e. above the threshold power). Additionally, individual nanoparticles have different size, facet shape and number of molecules which lead to slight variations in the threshold power and thus increase the spread of individual measurements. We now clarify the meaning of the error bars in the corresponding figure captions.

Reviewers' comments:

Reviewer #4 (Remarks to the Author):

The paper is much easier to follow now with the new format, and the argument is made more clear with the additional theory description added at the beginning.

The authors carefully discuss possible alternative interpretations and explain how they rule out possible artefacts. This is an important improvement. However, ruling out artifacts does in itself not strengthen the evidence for a claimed bond softening through the optomechanical spring effect being observed.

Overall, this reviewer is unconvinced that the authors observe the optomechanical spring effect. This reviewer simply does not see how the authors derive that the expected behavior from theory in Fig 5b is the underlying cause of the background increase seen in Fig 5a. And the claim otherwise rests almost entirely on the small deviation from linearity, almost within the noise, of the data in Fig 4e (grey points).

Following Carl Sagan's comment "Extraordinary claims require extraordinary evidence", this reviewer does not see that the work lives up to that principle.

However, perhaps the paper should be published anyway. Ultimately it is up to the authors to convince the scientific community. The work might inspire future experiments and it will be seen if the results and claims stand the test of time. This reviewer did its job, it is the authors who bear the responsibility if their conclusions are right or not.

Some additional comments for clarification and improvement:

1. This reviewer cannot think of a mechanism that could lead to a reversible saturation of the Raman signal. The author's argument depends on the idea that the saturation of the 1586 cm⁻¹ Raman peak is caused by bond softening. They argue why this saturation is not caused by other effects starting on line 276. Can the authors further elaborate on that aspect.

2. Fig. 5a, the black line is described as the "power-induced" spectral change, what does that mean? The caption makes it sound like the black curve plots the difference in the spectrum at high powers

compared to low powers, but it clearly wasn't calculated as $100I_i - I_i$ since that doesn't match the blue and red curves.

3. Line 312 claims that the condition for close-packed molecules giving rise to collective excitations is $\rho < d$. However, in equation (4), the exponent term is $e^{(\rho/\delta)}$ so it seems like the close-packed condition should be $\rho < \delta$. This reviewer does not see a definition for δ , is it related to d ? If it is then the claim in line 312 would make more sense.

4. Fig. 6a, the caption says that this plot shows $\Delta\omega_v$ for single-mode cavities. Is there a reason why multiple mode cavities are not considered? The argument for measurable bond-softening depends on the plasmonic cavity supporting multiple modes, yet only a single mode is considered when making this figure.

Reviewer comments

It is highly satisfying that reviewers #1 and #2 are delighted with the changes we have made to this manuscript, and urge for its publication. Reviewer #4 brings up several details, but most importantly they feel it is needed to directly show the shift of the SERS lines. We acknowledge their fair request. Having noted that pulsed excitation smears out the shifts, we undertook for this referee some additional CW experiments in nanocavities that have higher field enhancements. Indeed, we are now able to observe directly the reversible power-dependent shifts, and thus provide a suitably revised version of the manuscript that integrates these aspects together.

Reviewer #4:

The paper is much easier to follow now with the new format, and the argument is made more clear with the additional theory description added at the beginning.

> We appreciate the reviewer's suggestions, and that these improved the paper.

The authors carefully discuss possible alternative interpretations and explain how they rule out possible artefacts. This is an important improvement. However, ruling out artifacts does in itself not strengthen the evidence for a claimed bond softening through the optomechanical spring effect being observed. Overall, this reviewer is unconvinced that the authors observe the optomechanical spring effect. This reviewer simply does not see how the authors derive that the expected behavior from theory in Fig 5b is the underlying cause of the background increase seen in Fig 5a. And the claim otherwise rests almost entirely on the small deviation from linearity, almost within the noise, of the data in Fig 4e (grey points).

> As requested by the reviewer, we now provide direct observation of the linear spring shift (Fig.R1). A full discussion and comparison of pulsed and CW results is given in the main text.

Fig.R1: Fully-reversible spring shift seen for picocavity lines under CW excitation.

However, perhaps the paper should be published anyway. Ultimately it is up to the authors to convince the scientific community. The work might inspire future experiments and it will be seen if the results and claims stand the test of time. This reviewer did its job, it is the authors who bear the responsibility if their conclusions are right or not.

> This is useful. Indeed, one important consideration is that the theory itself does not stand on the experimental results, but is the straightforward (if sophisticated) application of current macroscopic cavity-QED theory to the plasmonic nanocavity system. This theory unambiguously says that large spring shifts will result in such systems, and this is an important breakthrough.

1. This reviewer cannot think of a mechanism that could lead to a reversible saturation of the Raman signal. The author's argument depends on the idea that the saturation of the 1586 cm⁻¹ Raman peak is caused by bond softening. They argue why this saturation is not caused by other effects starting on line 276. Can the authors further elaborate on that aspect.

> As described in the text, bond softening as predicted by the theory would shift and broaden the SERS lines. Different shifts would be found for different optical field strengths in different locations within the nanocavity, and for different times within the optical pulse. The net effect blurs the sharp SERS line, which thus reversibly saturates. We describe this and compare other possible mechanisms such as anharmonicity, providing careful estimates in the SI. This saturation is simply a description of what is observed: the sharp line disappears at high power, and returns at low power.

2. Fig. 5a, the black line is described as the “power-induced” spectral change, what does that mean? The caption makes it sound like the black curve plots the difference in the spectrum at high powers compared to low powers, but it clearly wasn’t calculated as $100I_i - I_i$ since that doesn’t match the blue and red curves.

> Because the spectra in Fig.5a average over molecules in different fields, we extract a spectrum which best represents the high power spectrum. We do this by subtracting from the high power $100I_i$ (red) curve a vertically scaled low power I_i (blue) spectrum that best removes the original SERS peaks. Hence, the black curve is $100I_i - cI_i$, now described in the caption. While this cannot be fit to the theory, it usefully shows the changes to the reader. It can be removed if the referee feels it misleads too much.

3. Line 312 claims that the condition for close-packed molecules giving rise to collective excitations is $\rho < d$. However, in equation (4), the exponent term is $e^{(\rho/\delta)}$ so it seems like the close-packed condition should be $\rho < \delta$. This reviewer does not see a definition for δ , is it related to d ? If it is then the claim in line 312 would make more sense.

> The reviewer really helpfully spots a definition we did not include, and we appreciate their careful reading. In equation (4), the exponent term arises in the denominator as $e^{(-\rho/\delta)}$, where the molecule spacing is ρ and $\delta = d/1.6$. This is presented in SI section S7, coming from analytic fits to the full theory. We thus replace this term suitably in equation (4).

4. Fig. 6a, the caption says that this plot shows $\Delta\omega_v$ for single-mode cavities. Is there a reason why multiple mode cavities are not considered? The argument for measurable bond-softening depends on the plasmonic cavity supporting multiple modes, yet only a single mode is considered when making this figure.

> The reviewer is right. In Fig. 6a we preferred to use a single cavity mode to put all the optomechanical cavities on the same footing when comparing the ratio of spring shift $\Delta\omega_v$ to effective optomechanical damping rate Γ_{tot} , which serves as the background of the contour plot. However, in order to emphasize the enhancement of this ratio in the plasmonic cavity, we show the results of the full plasmonic response in this case on the boxed foreground. We now mention this explicitly in the figure caption for clarity.

REVIEWER COMMENTS

Reviewer #5 (Remarks to the Author):

The manuscript by L. A. Jakob et al. reports optical spectroscopic data on plasmonic nanocavities containing Raman-active molecules, together with new theoretical developments treating the collective optomechanical interaction of an array of molecules with a single nanocavity. Overall, while the theoretical developments are truly interesting, and the data are also valuable due to the amount of nanocavities studied, I fail to see a clear connection between these two aspects of the manuscript, and would recommend publishing these studies either separately, or rewriting the manuscript in a less biased way (in particular changing the title to put the emphasis on the observation instead of the speculative interpretation). Also, it is quite frustrating (and alarming?) to see only statistically averaged data while the theory is developed for an individual nanocavity, and the spread in the individual data points visible in all figures raises concerns on the reality of the effect at the single cavity level.

I first address the Editor's question "to which extent the additional results added in response to reviewer #4 helps to corroborate the main claim of achieving the optomechanical spring effect." These results are now presented in Fig. 6, if I am not mistaken.

I see a huge problem with Fig. 6, which reports the effect of increasing laser power on the Raman spectrum of a so-called picocavity, i.e. a single-molecule event in SERS. Indeed, similar to the conflict with previous publications from the same group (Ref. 14 on claimed backaction amplification), we should compare Fig. 6 to the following published data from the same experimental group:

- Figs. 3A and 4B in Science 354, 726-729 (2016)

- Figs. 1c and 3d in Phys. Chem. Lett. 9, 24, 7146–7151 (2018)

- Fig. 1c in ACS Photonics 8, 10, 2868–2875 (2021)

These previous reports are of course just illustrations of a much larger body of data that exists on picocavities at Cambridge and elsewhere. They show that picocavity lines most often exhibit spectral

fluctuations during their lifetime (change in Raman shift and linewidth), the magnitude of which is similar to what is presented in Fig. 6. In fact, in the first publication listed above [Science 354, 726-729 (2016)], exactly the same experiment was performed (power sweep during picocavity emission), and while the lineshape was not shown, it can be inferred from the color plots that different lines show different and non-monotous shifts:

From this non-exhaustive review of published data on picocavities, it is very difficult to trust that Fig. 6 is really a demonstration of optical spring. Among the thousands and thousands of picocavity events that the authors have observed, it is expected that some of them will show the desired behavior, and possible reasons are numerous. The data of Fig. 6 look suspiciously non-representative.

In addition, I would like to highlight the major sources of concern I found elsewhere in the manuscript:

- a. Single NPoM in fig S20 does not show reversible saturation

If there truly is a reversible saturation of the Raman intensity it should be visible on individual NPoMs (which is what the model describes). But there is in fact **just one example** of individual NPoM data shown, in the SI in Fig. S20. Surprisingly (or not), this NPoM shows no saturation of Stokes intensity and no clear increase in background.

- b. Lack of consistency between experimental data and theoretical model in Fig. S19d

A careful look at Fig. S19d, which is the only piece of evidence provided in support of the reversibility of the effect, reveals all kinds of single-cavity behaviours (individual points), with both super- and sub-linear power dependence (positive and negative signals, respectively), and with all possible combinations of “damage” vs. “suppression”. For example, all data points in the upper half of the graph (with positive value of “suppression”) frontally contradict the model, which predicts that only saturation can occur. Data points on the lefthand half (negative value of “damage”) correspond to NPoMs showing a permanent decrease of Raman intensity with laser power, and this permanent change is not predicted by the optomechanical spring effect, yet it is NOT removed from the other figures of the main paper (at least I could find any statement that it was).

- c. Discrepancies between model prediction and experimental data (background interpretation)

Even if we were to trust the data analysis and believe that the x-renormalized and averaged spectra might be representative of individual NPoMs, there are clear discrepancies between model prediction and experimental data. One of them, seen in Figs. 4, 5 and also in Fig. S23, is the flat increase of background, both at lower **and higher (>1560 cm⁻¹) Raman shifts** compared to the main Raman peak. According to the model, collective modes occur only at lower Raman shifts, so it is wrong to associate the background to the collective modes (a key assumption of the paper)

Reviewer #5:

1. The manuscript by L. A. Jakob et al. reports optical spectroscopic data on plasmonic nanocavities containing Raman-active molecules, together with new theoretical developments treating the collective optomechanical interaction of an array of molecules with a single nanocavity. Overall, while the theoretical developments are truly interesting, and the data are also valuable due to the amount of nanocavities studied, I fail to see a clear connection between these two aspects of the manuscript, and would recommend publishing these studies either separately, or rewriting the manuscript in a less biased way (in particular changing the title to put the emphasis on the observation instead of the speculative interpretation).

> We thank the reviewer for emphasising both the theory interest and valuable data. Since we strongly believe that the theoretical models and experimental results substantially complement each other, we have opted to make changes to the manuscript to better reflect the connection of theory and experimental data. As suggested, we also change the title of the manuscript to better represent the experimental observation rather than just the conclusion of our model.

2. Also, it is quite frustrating (and alarming?) to see only statistically averaged data while the theory is developed for an individual nanocavity, and the spread in the individual data points visible in all figures raises concerns on the reality of the effect at the single cavity level.

> This comment suggests perhaps that the referee has a more theoretical-led approach, and thus is less aware of the reality of experimental SERS measurements on nanostructures. Compared to most works which show only a few select results, it is precisely the full data set explicitly presented here that gives confidence in our results (as positively noted by the previous referees). Groups around the world get vast inconsistencies in measuring nominally the same nanostructure. Our work endeavours to control as much about the precision architecture as possible, and yet still variations are seen. Our analysis provides a precise and consistent way to deal with this. However, we also see the key effects in raw data on individual nanostructures, as now explicitly added to the SI (section S13).

In our comprehensive measurements, SERS intensities fluctuate by up to one order of magnitude between NPoMs. The best way to meaningfully extract and understand new physical effects is to analyse a large dataset of many particles. A key experimental advance in this manuscript is showing how to better compare individual plasmonic nanostructures that give different SERS intensities using the concept of in-coupled laser intensity. The underlying reason for these deviations between NPoMs are nanoscale differences in facet and particle size and shape. We recently published a detailed study (<https://doi.org/10.1021/acsp Photonics.2c00116>) on the effect of facet shape demonstrating that plasmon resonances and thus SERS intensities vary drastically from particle to particle. Since control of these factors is fundamentally limited (by nanoparticle synthesis and reshaping), the only way to experimentally study SERS intensities is to record a large dataset on many particles combined with a robust way of analysing the data.

In this manuscript, we present a new data analysis metric, in-coupled laser intensity, that allows us to compare SERS intensities measured on different nanostructures despite their distinct differences. Essentially, we normalise the laser intensity for each NPoM by the SERS counts obtained at the lowest powers. This corrects for the efficiency of laser in-coupling into the gap and hence best estimates the actual intensity experienced by the molecules. We thus believe it is best to present averaged spectra (of NPoMs at the same in-coupled intensity) rather than individual spectra which can be similarly misleading since these need to be arbitrarily selected from a pool of many hundred particles. Nonetheless, we understand the reviewer's concern and thus add more raw data to the Supplementary Information to demonstrate the effect indeed exists for individual cavities. In addition, to clarify the advance made by introducing in-coupled intensities we present data for one laser wavelength with and without in-coupling correction in SI Figure S19.

3. I first address the Editor's question "to which extent the additional results added in response to reviewer #4 helps to corroborate the main claim of achieving the optomechanical spring effect."

These results are now presented in Fig. 6, if I am not mistaken. I see a huge problem with Fig. 6, which reports the effect of increasing laser power on the Raman spectrum of a so-called picocavity, i.e. a single-molecule event in SERS. Indeed, similar to the conflict with previous publications from the same group (Ref. 14 on claimed backaction amplification), we should compare Fig. 6 to the following published data from the same experimental group:

- Figs. 3A and 4B in Science 354, 726-729 (2016)

- Figs. 1c and 3d in Phys. Chem. Lett. 9, 24, 7146-7151 (2018)

- Fig. 1c in ACS Photonics 8, 10, 2868-2875 (2021)

These previous reports are of course just illustrations of a much larger body of data that exists on picocavities at Cambridge and elsewhere. They show that picocavity lines most often exhibit spectral fluctuations during their lifetime (change in Raman shift and linewidth), the magnitude of which is similar to what is presented in Fig. 6.

> Indeed the reviewer correctly points out that SERS lines in picocavities are often observed to exhibit time-dependent spectral shifts due to their single-molecule origin. However, we emphasise a crucial difference between these previous publications and the results presented in this manuscript: while spectral fluctuations are observed on timescales of a few to 10s of seconds, here we have made great experimental efforts to probe the power dependence on the millisecond timescale to avoid any slower changes in the spectrum. In our data [Fig. 6g], we clearly demonstrate that the SERS line reversibly returns to its original Raman shift at lowest laser power over the 0.08 s total observation time. We note that between these fast scans, a longer dead time occurs and indeed the picocavity line occasionally appears in a slightly shifted position. Since this consideration of fluctuating picocavity spectra clearly is very important, we now discuss this accordingly in the manuscript. We also show another 6 examples of the directly-observed picocavity shifts in the SI Fig.S30.

4. In fact, in the first publication listed above [Science 354, 726 (2016)], exactly the same experiment was performed (power sweep during picocavity emission), and while the lineshape was not shown, it can be inferred from the color plots that different lines show different and non-monotous shifts:

> Again, the important difference to experiments published in Science 354, 726 (2016) is the timescale on which the power sweep is carried out. In this previous publication, one power sweep was carried out over 100 seconds, incrementally changing the laser power for each spectrum. Here, we present a greatly advanced experimental technique (involving improved nanostructures, fast laser modulation using AOMs and specialised kinetic CCD readout) to conduct 50 power sweeps per second, four orders of magnitude faster than in the previous experiment. Due to the fluctuations of picocavity spectra pointed out above, this is the only way to meaningfully extract power-dependent lineshifts from picocavity spectra. We now highlight the necessity of this experimental upgrade in the main text.

5. From this non-exhaustive review of published data on picocavities, it is very difficult to trust that Fig. 6 is really a demonstration of optical spring. Among the thousands and thousands of picocavity events that the authors have observed, it is expected that some of them will show the desired behavior, and possible reasons are numerous. The data of Fig. 6 look suspiciously non-representative.

> Indeed, we observe many picocavity events in this experiment. The described optomechanical spring shift can be observed in the vast majority of picocavity time tracks, making the presented results representative for a typical picocavity. We accept the reviewer's suggestion that more data would be useful and add an additional figure in the SI showing time tracks for six different picocavities (Figure S30), note that almost all NPoM picocavities show this effect. We remind also that each picocavity

event is characterised by different vibrational lines depending on the exact position of the extracted Au atom [see J.Phys.Chem.Lett. 9, 7146 (2018)].

6. In addition, I would like to highlight the major sources of concern I found elsewhere in the manuscript: Single NPoM in fig S20 does not show reversible saturation: If there truly is a reversible saturation of the Raman intensity it should be visible on individual NPoMs (which is what the model describes). But there is in fact just one example of individual NPoM data shown, in the SI in Fig. S20. Surprisingly (or not), this NPoM shows no saturation of Stokes intensity and no clear increase in background.

> The data in previous SI Fig. S20 (now Fig. S24) cannot be directly compared to the main text since it was taken at a different laser wavelength (785 nm) and laser power where the SERS saturation effect is considerably weaker (explained below). Instead, wavelength and power were optimised to investigate the anti-Stokes signal as well as the Stokes.

Despite this much weaker effect, we still observe the effect of SERS saturation both in the individual spectra (Fig. S24a) as well as in the analysis of all particles (Fig. S24c). Additionally, a weak increase in background can be detected in the spectra in Fig. S24a. We agree with the reviewer that these finer changes are difficult to identify in individual particle spectra and therefore we present averaged spectra in the main text where the effect is much clearer. Finally, we emphasise that Fig. S24 actually supports our model of a spectral redistribution of SERS intensity since the total integrated SERS intensity (= sum of SERS lines + background, Fig. S24d) increases linearly even at high laser powers without saturation, ruling out molecular damage as the cause of SERS saturation.

To elaborate on the magnitude of the spring shift: there are two reasons why the saturation effect in the data presented in the SI is much weaker:

(i) at 785 nm laser wavelengths, our theory indeed predicts a weaker spring shift than at 658 nm due to the positions of plasmonic modes (see Fig. 2a). Hence, the threshold power to markedly change the spectrum will be higher.

(ii) to record the much weaker anti-Stokes signal, integration times had to be extended fourfold compared to the measurements in the main text. Unfortunately, this increases damage to the nanostructures and thus the maximum power in the experiment had to be reduced threefold.

Hence, the intensity regime where the optical spring shift becomes strongly visible is not reached in this experiment. We now note this important clarification in section S15 of the SI to avoid confusion of the reader.

7. Lack of consistency between experimental data and theoretical model in Fig. S19d

A careful look at Fig. S19d, which is the only piece of evidence provided in support of the reversibility of the effect, reveals all kinds of single-cavity behaviours (individual points), with both super- and sublinear power dependence (positive and negative signals, respectively), and with all possible combinations of “damage” vs. “suppression”. For example, all data points in the upper half of the graph (with positive value of “suppression”) frontally contradict the model, which predicts that only saturation can occur. Data points on the lefthand half (negative value of “damage”) correspond to NPoMs showing a permanent decrease of Raman intensity with laser power, and this permanent change is not predicted by the optomechanical spring effect, yet it is NOT removed from the other figures of the main paper (at least I could find any statement that it was).

> Figure S19 (now Fig. S23) summarises two different experiments to demonstrate the reversibility of the saturation effect. Both independently verify that the SERS saturation is reversible and different from molecular damage. Firstly, we carry out two full power sweeps on each nanoparticle and plot the results of many particles on top of each other (with averaged data as filled points). As is visible very clearly, both power series overlap very well with only a small offset in count rate due to minor

damage. The saturation effect is similarly observed in both power sweeps. Secondly, we alternately record spectra at low and high laser intensity for multiple cycles. In this experiment, it is not possible to account for each nanoparticle's in-coupling efficiency since the two laser powers are fixed. Therefore, fluctuations between individual NPoMs cannot be accounted for and lead to outliers in Fig. S23d not showing any suppression (for particles with low in-coupling efficiency) or even an increase for higher power (possible in individual cases due to laser-induced, nanoscale changes to the cavity). One should not view these outliers as 'frontally contradicting the model', but rather as experimental complications from single nanostructure spectroscopy where the actual atomic-scale precision required is not (yet) available. Since the outliers can be misleading to the eye and the vast majority of data points overlap in the shaded areas of the plot, we suggest looking at the histograms at the sides of Fig. S23d. Then, the evidence for the reproducibility of the effect becomes quite clear. This is now more clearly explained in the figure caption. Further, we better explain the two experiments carried out to demonstrate reproducibility in SI section S14.

8. Discrepancies between model prediction and experimental data (background interpretation)

Even if we were to trust the data analysis and believe that the x-normalized and averaged spectra might be representative of individual NPoMs, there are clear discrepancies between model prediction and experimental data. One of them, seen in Figs. 4, 5 and also in Fig. S23, is the flat increase of background, both at lower and higher (>1560 cm^{-1}) Raman shifts compared to the main Raman peak. According to the model, collective modes occur only at lower Raman shifts, so it is wrong to associate the background to the collective modes (a key assumption of the paper)

> Indeed, the reviewer rightly points out that there also is a rise in the background on the high energy side of the Raman peak – something omitted from the text to avoid confusing readers further. However, the increase of the background is much stronger on the lower energy side (see Figure R1 below, also now improved Fig. 5a) consistent with our proposed theory of an optomechanical spring shift. We suggest that an additional effect leads to a background increase at all Stokes shifts: the illumination of the nanostructures with a pulsed laser transiently increases the non-equilibrium 'temperature' by many hundred degrees (see Fig. S25b). These excitations can also affect the SERS background obtained from electronic Raman scattering (ERS), which on the Stokes side of the spectrum leads to a constant contribution. This effect is completely independent from the molecular SERS signal shifting and broadening due to the optical spring effect. We appreciate the reviewer's point which helps us clarify this aspect of the paper with improved Fig. 5a and SI section S15.

Fig. R1: Background increase at high laser power. SERS background at high laser power increases on both the high- and low-wavenumber side of the vibrational line. This change in background can be attributed to a constant uplift across the spectrum, due to an increase in electronic Raman scattering from transient non-equilibrium 'temperature' increases, as well as the redistribution of vibrational energy by the optical spring shift towards lower wavenumbers.

REVIEWER COMMENTS

Reviewer #5 (Remarks to the Author):

I thank the authors for replying to my questions and for showing more experimental data in the revised version, which somewhat improves the transparency of the whole study. In particular, the few single-particle datasets that the authors accepted to show confirmed my expectations that the dominant effect witnessed under pulsed measurement is an abrupt change of regime in the Raman spectrum at some power, at odds with the optomechanical spring effect. The additional picocavity data are very nice, but they are not helping in supporting the interpretation of the pulsed data in terms of optomechanical spring effect – on the contrary, as explained below, I found it impossible to reconcile the magnitude of the effect in the two different experiments using the equations provided by the authors.

At this stage, after a very careful study of the material provided by the authors, I conclude that the observations under pulsed excitation do not support the theoretical model and that their interpretation should be revised and left more open. On the other hand, the picocavity data seem more reproducible and show clear shifts. Yet, given that extremely strong vibrational pumping was reported by the same group in picocavities, the observed downshifts seem to agree well with the numbers predicted in Section S9, so that anharmonicities should be considered more thoroughly in the context of picocavities as a possible explanation of the power-dependent shifts.

Regarding the replies to the 8 points:

1) The new title still conveys the impression that a “giant optomechanical shift” was demonstrated. I disagree with this statement for all the reasons stated across this report and the previous one.

2) The authors agreed to add single-particle data, which I find to be the minimal requirement for transparency in such an experimental study. The new data are useful, and I include below my analysis of them. I conclude that single nanocavities tend to present an abrupt change of spectrum at a given pump power (dashed red vertical lines), an effect very well known among researchers working with SECARS and other pulsed SERS techniques. It is likely that the smooth saturation effect only emerges after averaging of 100's of datasets, as would be expected.

3-5) I thank the authors for clarifying the speed of the power sweeps and for including more picocavity measurements in Fig. S30, showing power-dependent shift ranging from 5 to 20 cm^{-1} .

First, I find it surprising that the authors do not even try to connect the new data on picocavities directly to the model predictions for single molecule, but instead propose a hard-to-follow comparison with the pulsed measurement in different nanocavities. I tried to reproduce their calculations and find a very different result. If I use eq. (5) and consider 7-time larger EF and 100 times less molecules (what is mentioned in the text), the optical spring shift should be half smaller for SPARK picocavities than for nanocavities under same laser intensity, whereas it is claimed to be $1e2$ to $1e3$ times larger (since a spring shift of up to 20 cm^{-1} is seen at $3e-4$ times less power). Something may be wrong with the interpretation.

Second, from Fig. S12, for a single molecule ideally positioned in a nanocavity, the spring shift for $1 \mu\text{W}/\mu\text{m}^2$ incident power is predicted to be on the order of $1e-7 \text{ meV}$. In Fig. 6, the SPARK cavities are driven with up to $300 \mu\text{W}$ (power density not given). Assuming a sub-micron spot size let's say $1 \text{ mW}/\mu\text{m}^2$, which would, for a standard nanocavity, lead to the prediction of $1e-4 \text{ meV}$ spring shift. The observed shifts are up to $20 \text{ cm}^{-1} = 2.5 \text{ meV}$, i.e. at least **$1e4$ larger**. The increase in field enhancement for a picocavity is not known precisely, but for a single molecule to outshine 100 others it suffices that EF^4 is increased by ~ 100 , so we can assume that EF^2 is increased by a factor of order 10 (and correspondingly the spring shift if we use eq. 5). It still comes short of $1e4$ needed to match the model prediction.

In the end, the new data and their unclear analysis further weaken my trust in the main claim of the manuscript.

6) No further comments.

7) On the question of “suppression” vs. “damage”, I do understand what is plotted in Fig. S23 and thank the authors for confirming my understanding. Still, from these plots remain suspicious about the strong statements on reversibility that the authors make in the text and in their reply. I propose that they show the averaged spectrum at low-power (i) before first power sweep; (ii) after first power-sweep; (iii) after second power-sweep. It would allow to assess how well these 3 spectra overlap.

A further analysis could be done by post-selecting among all data only the nanocavities showing no change in low-power spectrum after sweep 1 and 2, and perform a restricted analysis on these cavities, in order to evaluate how much of the effect remains after irreversible changes have been removed. After all, the histograms in Fig S23d show that about half of the effect size is irreversible (~20% vs. 45%).

8) “However, the increase of the background is much stronger on the lower energy side (see Figure R1 below, also now improved Fig. 5a)”

First, in Fig. S27, we can appreciate that for TPT molecules the relative increase in background at high vs. low power is similar *below* and *above* the highest frequency Raman peak. It suggests the difference in background increase observed for BPT may be just an accident and has nothing to do with the optomechanical spring effect. Second, if we look at the relative increase of background emission, we find it to be very similar on both sides of the high-frequency Raman peak (both for BPT and TPT), as shown below:

Therefore, I maintain my affirmation that the data are at odds with the optomechanical spring effect predictions.

To finish, another few questions that emerged while going a second time through the SI:

9) Fig. S6 : how many molecules are considered in this simulation? Only two, correct? I did not find the information clearly stated for the entire figure.

10) Fig. S10a: We see that resonance shift and linewidth broadening are predicted to be of similar magnitude. Where lies the difference with Fig. S14 which predicts widely different values for these two quantities?

11) Eq. S31, S32: It is known that for a single molecule in a single-mode cavity, the optical spring should scale as g^2/κ^2 (in Doppler limit), which translates as a scaling proportional to $N_m * EF^4 * I_L$ in terms of field enhancement factor EF and laser intensity I_L . In contrast, Eq. S32 presents a scaling as $N_m * EF^2 * I_L$. Could the authors comment on this discrepancy? Is the other EF^2 factor hidden somewhere in the equation? If so, could it be made explicit? If not, shouldn't the model be discarded in case it fails to recover the standard theory prediction?

Reviewer #6 (Remarks to the Author):

I was asked to assess the manuscript “Giant optomechanical spring effect in plasmonic nano- and picocavities probed by surface-enhanced Raman scattering” specifically regarding the following questions:

a) Point [2] which relates to the experimental demonstration of the optomechanical spring effect and the evidence that can be drawn by analyzing the averaged picocavity data. Our question is: Is the authors’ reply to point 2, where they explain the method to extract average data, scientifically sound? Does it give a solid tool to interpret data and demonstrate the optomechanical spring effect?

b) Points [3-5] that relate to the information that can be gathered by analyzing single NPoMs behavior. Is the discrepancy between the model and the theory enough to conclude that the authors experimentally probed the optomechanical spring effect? Does the additional single cavity data really corroborate the fact that the optomechanical spring effect is not substantiated?

c) Points [9-11]. Here we have additional technical concerns that ask the authors to add more data. Are these additional requests determinants to substantiate the final claims or are they minor technical issues that do not compromise the scientific substantiation of the result?

My responses are as follows:

A. Linear normalization using low power response is logically sound and removes most of the discrepancy between the individual nanocavity datasets (Fig S19). (Side note: picocavity data is not averaged. It cannot be averaged in the same way because the picocavity spectra vary). Averaging data from individual device measurements after this normalization is a sound approach. It clearly reveals downward trend for the main peak area with increasing power and initial upward trend in the “background” area. However data fig 4g,h,i, 5d also shows downward trend in the background at the highest powers, whereas the theory in fig 5 does not. While it is possible that an additional process accounts for it, it does not seem to be discussed in the paper.

Clearly, ability to analyze multiple datasets and reduce them to the average, while also showing qualitatively similar trends in individual data makes the measurement stronger.

Furthermore, if, literally, an “abrupt change of spectrum” were to occur at a given power, it seems this would result in a discontinuity in the signal data vs. incident power, rather than the deviation from linear scaling of signal vs. power, which is being experimentally observed. That said, I lack the pulsed-SERS expertise to further comment on the Rev#5 reference to “... an abrupt change of spectrum at a given pump power (dashed red vertical lines), an effect very well known among researchers working with

SECARS and other pulsed SERS techniques.” While the individual data presented and the averaged data are not incompatible with linear scaling turning to a power law with exponent <1 , as drawn by the Rev#5, it is unclear to me when such a model might be applicable.

I conclude that the data presented and analyzed is compatible with the softening hypothesis of the paper, as is also stated by the authors. However, taken by themselves without considering the picocavity data, nanocavity data are insufficient to conclusively demonstrate the optomechanical spring nature of the observed effects. Due to the inevitable smearing in the pulsed observation, and lack of temporal resolution in the current observation modality it is not possible to obtain the type of spectral shape information that might unequivocally and convincingly confirm the optical spring explanation for the observed saturation.

To summarize, averaging method is scientifically sound and averaged data from many devices strongly increases the confidence in experimental observations. It is thus a solid tool for data interpretation. The nanocavity data is compatible with the optomechanical spring effect, but taken strictly by itself, is insufficient to unequivocally demonstrate it.

B. Picocavity data, presumed from single molecules, is used to provide critical support to the optical spring hypothesis. It clearly shows repeatable and reversible red tuning and broadening with increased power. It is stated to be reproducible for all bright-signal picocavities, and the majority of all picocavities, though the extent of tuning is variable. These observations appear sound.

The authors claim an order-of-magnitude consistency between picocavity and nanocavity tuning. However, similar to Rev#5, I fail to reproduce this analysis. For example, Eq.S32 appears to relate Δ_1 and Δ_N and indicate scaling with EF and intensity. The $EF^2 \cdot I$ ratio is $49 \cdot 3e-4 = 0.0147$ while N is stated ~ 100 elsewhere and η_1 is stated ~ 0.13 . Thus, with the effective number of bright-mode molecules ~ 13 , the ratio is $\sim 1e-3$ rather than the $\sim 1/50$ as stated by the authors.

In my view picocavity data is critical to experimentally support the optical spring hypothesis. The picocavity shift must be shown to be at least within an order-of-magnitude of a plausible and clearly explained theoretical estimate. Unfortunately, because of this deficiency, I must conclude that the theory of the optical spring effect is not currently sufficiently supported by experimental observations to unequivocally claim the observation of the theorized effect. However, if picocavity observations can be clearly shown to be consistent with theory, the overall evidence would be sufficient to claim the experimental observation of the optical spring effect.

C. (9) – minor point; (10) – this would be useful to explain, and probably straightforward to do so - presumably plasmonic modes cannot be neglected when calculating broadening? (11) – I do not

consider this a determinant. If I understand the question correctly, qualitatively, for an optical cavity Q^2 scaling comes from both field and lifetime enhancements, while here it is different because G is not dominated by a single mode.

Reviewer #5:

(1) In particular, the few single-particle datasets that the authors accepted to show confirmed my expectations that the dominant effect witnessed under pulsed measurement is an abrupt change of regime in the Raman spectrum at some power, at odds with the optomechanical spring effect.

> We indeed thank the reviewer for the careful analysis of our data. However, we do not agree with their interpretation of an ‘abrupt change of regime’ as discussed in point (6) in more detail. As stated in our article and previous response, we examine hundreds of experiments and present carefully averaged data demonstrating the experiment is indeed in good agreement with the optomechanical model. To give more evidence that the same saturation effect is reproduced by individual particles, we include now a fit with the optomechanical model in Figs. S20-22 and below in Fig. R1.

(2) The additional picocavity data are very nice, but [...] I found it impossible to reconcile the magnitude of the effect in the two different experiments using the equations provided by the authors.

> We discuss this carefully below and sort out the inconsistencies. Previous reviewers asked if CW pumping also showed the shifts (which can only be observed in picocavity data), and which was then added. We believe the agreement of the two different experiments actually builds the more convincing case, which according to our estimates are fully consistent.

(3) I conclude that the observations under pulsed excitation do not support the theoretical model and that their interpretation should be revised and left more open.

> We point out that we are already very careful about interpreting the results in the manuscript. For example, in the abstract we only state that ‘the theoretical results are *consistent* with the experimentally-observed strongly non-linear behaviour’. Further, we report ‘*indications* of a vibrational frequency shift associated with the optical spring effect’ and conclude that the ‘physical picture is *commensurate* with experimental observations’. Finally, we now add the following sentence to the conclusion of the manuscript: “Independently, the two experiments cannot unambiguously confirm the existence of the proposed optical spring shift, however both are in good agreement with the optomechanical model developed here.”

(4) On the other hand, the picocavity data seem more reproducible and show clear shifts. Yet, given that extremely strong vibrational pumping was reported by the same group in picocavities, the observed downshifts seem to agree well with the numbers predicted in Section S9, so that anharmonicities should be considered more thoroughly in the context of picocavities as a possible explanation of the power-dependent shifts.

> The reviewer raises a valuable point. We considered anharmonicities in Section S9 for pulsed excitation in NPoMs and show in Fig. S16 that 10 cm^{-1} shifts require $10^7\text{ }\mu\text{W}/\mu\text{m}^2$ (as shift proportional to laser intensity). With $I_l(\text{CW})/I_l(\text{pulsed}) = 3 \times 10^{-4}$ and $EF_{\text{SPARK,pico}}/EF_{\text{NPoM,nano}} \approx 20$ (see point 7), we expect the anharmonicity shift in our CW experiment to be more than ten-fold smaller than the shift observed. While anharmonic effects cannot be fully discarded based on this (simplistic) approximation, future work on anharmonicities in picocavities must be developed but is beyond the scope of this manuscript. To bring this to the attention of the reader, we added a paragraph at the end of the experimental section in the main text.

(5) The new title still conveys the impression that a “giant optomechanical shift” was demonstrated. I disagree with this statement for all the reasons stated across this report and the previous one.

> As noted above, our data is consistent with this title. We emphasise consistency not demonstration, and the title equally refers to the theory advances presented here. In particular, Fig. 7b exactly compares the shift per photon, and shows it is at least 10^3 larger than any other system.

(6) [...] The new data are useful, and I include below my analysis of them. I conclude that single nanocavities tend to present an abrupt change of spectrum at a given pump power (dashed red vertical lines), an effect very well known among researchers working with SECARS and other pulsed SERS techniques. It is likely that the smooth saturation effect only emerges after averaging of 100's of datasets, as would be expected.

> We thank the reviewer for taking time to perform a detailed analysis on our raw data. In their explanation (although not stated explicitly), it is permanent damage that reduces the signal at higher powers. However in complete contrast, our data is shown to be reversible. In addition, the total SERS emission (peaks plus backgrounds, Fig. R1 and revised S20-22) remains linear with power, as expected in our model which redistributes the vibrational energy optomechanically (but not in their damage explanation). The reviewer's plot implies a change of regime from linear scaling to a sublinear power law above a threshold laser intensity, but they give no explanation. Our optomechanical model shows why saturation is observed and leads to an excellent fit to the data with only two free parameters (blue line, Fig. R1), now also included for individual particles in Figs. S20-22. This good agreement with the experimental data clearly supports our model.

Fig. R1: Integrated SERS signal from individual NPOMs. The area of the 1586 cm^{-1} line (blue dots) is reproduced well by a fit of the optomechanical model (blue line), while the sum of all SERS signals (open circles) retains linear scaling (black line) after exceeding the power threshold for SERS saturation.

(7) First, I find it surprising that the authors do not even try to connect the new data on picocavities directly to the model predictions for single molecule, but instead propose a hard-to-follow comparison with the pulsed measurement in different nanocavities. I tried to reproduce their calculations and find a very different result. If I use eq. (5) and consider 7-time larger EF and 100 times less molecules (what is mentioned in the text), the optical spring shift should be half smaller for SPARK picocavities than for nanocavities under same laser intensity, whereas it is claimed to be 1e2 to 1e3 times larger (since a spring shift of up to 20 cm⁻¹ is seen at 3e-4 times less power). Something may be wrong with the interpretation.

> Full modelling of optical response in SPARK nanostructures, including a gold adatom picocavity, is beyond current capabilities. Therefore, a simpler approach was chosen estimating the order of magnitude of spring shifts expected. We appreciate that the detail given in the main text was insufficient to reproduce our calculations and therefore now describe it more carefully. Perhaps this lack of information in the previous version led to a few incorrect assumptions and oversights by the reviewer. Firstly, the reviewer misses out the additional field enhancement by the picocavity. We agree with the reviewer's estimate that the picocavity increases EF² by ~10. Secondly, we obtain from simulations in Fig. 2b that the spring shift of 100 molecules in the centre of the facet is only about 12 times larger than a single molecule in the centre (as noted in the main text), giving another order of magnitude difference from the estimate by the reviewer. The full analytical formula (Eq. S32) takes this into account with the factor η_1 describing the effective coupling to the collective Raman bright mode, where here, $\eta_1 = 0.12$. To simplify our results, we then included this factor in the proportionality constants in equations 4, 5 and S33, S34. We thank the reviewer for pointing out how this can be misleading to the reader and now explicitly include η_1 in all equations, adapting the proportionality factors accordingly. With these corrections, our estimates predict the correct order of magnitude for the spring shift in SPARK picocavities with exact numbers now in the main text.

For full transparency, we include in Table R1 all estimates made for our calculations based on eq. (5):

$$\Delta\omega_v \propto \eta_1 N_m \frac{(EF \cdot R_v)^2}{\omega_v} I_l$$

Table R1. Estimations for the comparison of spring shifts observed in the two experimental configurations

Physical quantity	Ratio CW SPARK picocavity / pulsed NPoM nanocavity
Nanostructure field enhancement	$\frac{EF_{\text{SPARK}}}{EF_{\text{NPoM}}} = 7$ (from experimental SERS counts)
Picocavity field enhancement	$\frac{EF_{\text{picocavity}}}{EF_{\text{nanocavity}}} = 3$ (from experimental SERS counts)
Effective molecule number	$\frac{1}{\eta_1 N_m} = \frac{1}{12}$ (from Fig. 2b)
Laser intensity	$\frac{I_l(\text{CW})}{I_l(\text{pulsed})} = 3 \cdot 10^{-4}$

With these approximations, we find that $\Delta\omega_{\text{NPoM}}/\Delta\omega_{\text{SPARK}} \approx 90$. In our experiments we observe a shift >250 cm⁻¹ with NPoM under pulsed illumination compared to ~5 cm⁻¹ in SPARK picocavities, giving a ratio of ~50. With the limitations of the model for SPARK picocavities, this can be considered good agreement, certainly within an order of magnitude, supporting the theory of an optomechanical spring shift.

(8) Second, from Fig. S12, for a single molecule ideally positioned in a nanocavity, the spring shift for $1 \mu\text{W}/\mu\text{m}^2$ incident power is predicted to be on the order of $1\text{e-}7$ meV. In Fig. 6, the SPARK cavities are driven with up to $300 \mu\text{W}$ (power density not given). Assuming a submicron spot size let's say $1 \text{mW}/\mu\text{m}^2$, which would, for a standard nanocavity, lead to the prediction of $1\text{e-}4$ meV spring shift. The observed shifts are up to $20 \text{cm}^{-1} = 2.5 \text{meV}$, i.e. at least $1\text{e}4$ larger. The increase in field enhancement for a picocavity is not known precisely, but for a single molecule to outshine 100 others it suffices that EF^4 is increased by ~ 100 , so we can assume that EF^2 is increased by a factor of order 10 (and correspondingly the spring shift if we use eq. 5). It still comes short of $1\text{e}4$ needed to match the model prediction. In the end, the new data and their unclear analysis further weaken my trust in the main claim of the manuscript.

> Figure S12 explicitly simulates the optomechanical parameters for the NPoM structure. Since the plasmonic modes and Greens function for a SPARK construct are drastically different, this simulation cannot be applied to our picocavity data in SPARKs. Unfortunately, there is no detailed understanding yet of plasmonic modes in SPARKs (since darkfield scattering is obscured by reflections from the silica microlens). It is thus not yet possible to carry out detailed simulations of the optomechanical parameters for our picocavity data. Our efforts to estimate the expected spring shift for such cavities (see point 7) thus rely on approximate field enhancements estimated from the SERS intensity. We thank the reviewer for pointing out these difficulties and now clearly comment on these limitations in the main text.

(9) On the question of “suppression” vs. “damage”, ...show the averaged spectrum at low-power (i) before first power sweep; (ii) after first power-sweep; (iii) after second power-sweep. It would allow to assess how well these 3 spectra overlap. A further analysis could be done by post-selecting among all data only the nanocavities showing no change in low-power spectrum after sweep 1 and 2, and perform a restricted analysis on these cavities, in order to evaluate how much of the effect remains after irreversible changes have been removed. After all, the histograms in Fig S23d show that about half of the effect size is irreversible ($\sim 20\%$ vs. 45%).

> We specifically take the advice of reviewer #6 here that our data analysis is robust and well-founded. We already discard all data that show significant evidence of drastic damage from such permanent changes.

(10) “However, the increase of the background is much stronger on the lower energy side (see Figure R1 below, also now improved Fig. 5a)” First, in Fig. S27, we can appreciate that for TPT molecules the relative increase in background at high vs. low power is similar below and above the highest frequency Raman peak. It suggests the difference in background increase observed for BPT may be just an accident and has nothing to do with the optomechanical spring effect. Second, if we look at the relative increase of background emission, we find it to be very similar on both sides of the high-frequency Raman peak (both for BPT and TPT), as shown below. Therefore, I maintain my affirmation that the data are at odds with the optomechanical spring effect predictions.

> The reviewer here again shows the increase of the background without the subtraction of a broadband background from electronic Raman scattering that we previously discussed with them. Once this analysis is carried out carefully, we find that, like BPT, the TPT SERS emission becomes stronger on the low-energy side. Thus, we conclude that the optomechanical spring effect is indeed seen in both TPT and BPT.

(11) Fig. S6 : how many molecules are considered in this simulation? Only two, correct? I did not find the information clearly stated for the entire figure.

> The reviewer is correct that in Fig. S6 two molecules are considered. We now clarify this explicitly in the figure caption.

(12) Fig. S10a: We see that resonance shift and linewidth broadening are predicted to be of similar magnitude. Where lies the difference with Fig. S14 which predicts widely different values for these two quantities?

> Again, we thank the reviewer for their careful reading. As explicitly noted for Eq. S39 (to which Fig. S14 refers), this is a simplistic prediction only in the very low wavenumber limit. By comparison, our experimental data does show similar shifts and broadening, as matching Fig. S10a. We thus carefully clarify the caption to Fig. S14 to ensure this is clear.

(13) Eq. S31, S32: It is known that for a single molecule in a single-mode cavity, the optical spring should scale as g^2/κ^2 (in Doppler limit), which translates as a scaling proportional to $N_m * EF^4 * I_L$ in terms of field enhancement factor EF and laser intensity I_L . In contrast, Eq. S32 presents a scaling as $N_m * EF^2 * I_L$. Could the authors comment on this discrepancy? Is the other EF^2 factor hidden somewhere in the equation? If so, could it be made explicit? If not, shouldn't the model be discarded in case it fails to recover the standard theory prediction?

> As the reviewer notes, the scaling with EF^4 is obtained for a single-mode cavity. We show in this manuscript that it is indeed very important to take the full multimode plasmonic cavity into account to reproduce the experimental observations. In the improved cavity multimode description, the optical spring shift scales with the square of the field enhancement, EF , and the real part of the Green's function, $Re(G)$, as: $EF^2 * Re(G)$. For a single-mode cavity $G \propto EF^2$ (due to reciprocity) and thus we recover the EF^4 scaling. However, in our system G is dominated by the interaction with the mirror charges, as described in section S3.1. Thus, G is approximately given by equation S25, which depends on the material properties and size of the gap instead of on EF^2 . We add a comment on this to the SI at the end of page 22.

Reviewer #6:

(1) Linear normalization using low power response is logically sound and removes most of the discrepancy between the individual nanocavity datasets (Fig S19). (Side note: picocavity data is not averaged. It cannot be averaged in the same way because the picocavity spectra vary). Averaging data from individual device measurements after this normalization is a sound approach. It clearly reveals downward trend for the main peak area with increasing power and initial upward trend in the "background" area. However data fig 4g,h,i, 5d also shows downward trend in the background at the highest powers, whereas the theory in fig 5 does not. While it is possible that an additional process accounts for it, it does not seem to be discussed in the paper.

Clearly, ability to analyze multiple datasets and reduce them to the average, while also showing qualitatively similar trends in individual data makes the measurement stronger.

> The reviewer indeed picks out an additional process at the highest power seen in the background. We believe this is damage to the molecules, and it is briefly mentioned. We believe damage can occur through the bond softening process discussed here, but cannot yet prove this. As suggested, we now mention this in the conclusions.

(2) Furthermore, if, literally, an "abrupt change of spectrum" were to occur at a given power, it seems this would result in a discontinuity in the signal data vs. incident power, rather than the deviation from linear scaling of signal vs. power, which is being experimentally observed. That said, I lack the pulsed-SERS expertise to further comment on the Rev#5 reference to "... an abrupt change of spectrum at a given pump power (dashed red vertical lines), an effect very well known among researchers working with SECARS and other pulsed SERS techniques." While the individual data presented and the averaged data are not incompatible with linear scaling turning to a power law with exponent <1 , as drawn by the Rev#5, it is unclear to me when such a model might be applicable.

> We comment on this issue in our response (6) to reviewer #5. Indeed, we now provide even more convincing evidence that the single particle data is consistent with an optomechanical spring shift.

(3) I conclude that the data presented and analyzed is compatible with the softening hypothesis of the paper, as is also stated by the authors. However, taken by themselves without considering the picocavity data, nanocavity data are insufficient to conclusively demonstrate the optomechanical spring nature of the observed effects. Due to the inevitable smearing in the pulsed observation, and lack of temporal resolution in the current observation modality it is not possible to obtain the type of spectral shape information that might unequivocally and convincingly confirm the optical spring explanation for the observed saturation.

To summarize, averaging method is scientifically sound and averaged data from many devices strongly increases the confidence in experimental observations. It is thus a solid tool for data interpretation. The nanocavity data is compatible with the optomechanical spring effect, but taken strictly by itself, is insufficient to unequivocally demonstrate it.

> We thank the reviewer for their positive assessment of our methods. We added a sentence in the conclusions of the manuscript to highlight that the experimental observation of SERS saturation alone is not unambiguous confirmation of the spring shift.

(4) Picocavity data, presumed from single molecules, is used to provide critical support to the optical spring hypothesis. It clearly shows repeatable and reversible red tuning and broadening with increased power. It is stated to be reproducible for all bright-signal picocavities, and the majority of all picocavities, though the extent of tuning is variable. These observations appear sound.

The authors claim an order-of-magnitude consistency between picocavity and nanocavity tuning. However, similar to Rev#5, I fail to reproduce this analysis. For example, Eq.S32 appears to relate Δ_1 and Δ_N and indicate scaling with EF and intensity. The $EF^2 \cdot I$ ratio is $49 \cdot 3e-4 = 0.0147$ while N is stated ~ 100 elsewhere and η_1 is stated ~ 0.13 . Thus, with the effective number of bright-mode molecules ~ 13 , the ratio is $\sim 1e-3$ rather than the $\sim 1/50$ as stated by the authors.

In my view picocavity data is critical to experimentally support the optical spring hypothesis. The picocavity shift must be shown to be at least within an order-of-magnitude of a plausible and clearly explained theoretical estimate. Unfortunately, because of this deficiency, I must conclude that the theory of the optical spring effect is not currently sufficiently supported by experimental observations to unequivocally claim the observation of the theorized effect. However, if picocavity observations can be clearly shown to be consistent with theory, the overall evidence would be sufficient to claim the experimental observation of the optical spring effect.

> As suggested by the reviewer, we demonstrate in our response 7 to reviewer #5 in detail that the optomechanical theory indeed agrees well (within the limitations of this simplified model) with the picocavity shifts observed. This is also reflected in a more transparent explanation of our estimates in the main text. With this agreement of modelling and experiment, we believe, like the reviewer, that the picocavity data provides valuable evidence for the optical spring effect.

(5) (9) – minor point; (10) – this would be useful to explain, and probably straightforward to do so - presumably plasmonic modes cannot be neglected when calculating broadening? (11) – I do not consider this a determinant. If I understand the question correctly, qualitatively, for an optical cavity Q^2 scaling comes from both field and lifetime enhancements, while here it is different because G is not dominated by a single mode.

> We responded as suggested, as indicated in the responses (12) and (13) to reviewer #5.

REVIEWERS' COMMENTS

Reviewer #6 (Remarks to the Author):

I have previously stated that: " The picocavity shift must be shown to be at least within an order-of-magnitude of a plausible and clearly explained theoretical estimate. ... if picocavity observations can be clearly shown to be consistent with theory, the overall evidence would be sufficient to claim the experimental observation of the optical spring effect."

I believe that my not considering the additional factor $(3x)^2$ enhancement, now explicitly reported and supported by Raman counts, explains my previous inability to reproduce the estimates. As a result, the experimental and estimated shifts indeed seem to differ by less than a factor of 2, showing reasonable consistency between the approximate model and observation.

I conclude that the overall evidence, as now reported, is sufficient to claim the experimental observation of the optical spring effect.

I also note that the authors sufficiently addressed all the other concerns, and therefore recommend publication.

Reviewer #6:

I have previously stated that: "The picocavity shift must be shown to be at least within an order-of-magnitude of a plausible and clearly explained theoretical estimate. ... if picocavity observations can be clearly shown to be consistent with theory, the overall evidence would be sufficient to claim the experimental observation of the optical spring effect." I believe that my not considering the additional factor $(3x)^2$ enhancement, now explicitly reported and supported by Raman counts, explains my previous inability to reproduce the estimates. As a result, the experimental and estimated shifts indeed seem to differ by less than a factor of 2, showing reasonable consistency between the approximate model and observation. I conclude that the overall evidence, as now reported, is sufficient to claim the experimental observation of the optical spring effect. I also note that the authors sufficiently addressed all the other concerns, and therefore recommend publication.

> We are delighted that the reviewer is convinced by our arguments and thank them for carefully reproducing our calculations.